# DP-HyPO: An Adaptive Private Hyperparameter Optimization Framework

**Hua Wang**
Department of Statistics and Data Science
University of Pennsylvania
Philadelphia, PA 19104
`wanghua@wharton.upenn.edu`

**Sheng Gao**
Department of Statistics and Data Science
University of Pennsylvania
Philadelphia, PA 19104
`shenggao@wharton.upenn.edu`

**Huanyu Zhang**
Meta Platforms, Inc.
New York, NY 10003
`huanyuzhang@meta.com`

**Weijie J. Su**
Department of Statistics and Data Science
University of Pennsylvania
Philadelphia, PA 19104
`suw@wharton.upenn.edu`

**Milan Shen**
Meta Platforms, Inc.
Menlo Park, CA 94025
`milanshen@gmail.com`

## Abstract

Hyperparameter optimization, also known as hyperparameter tuning, is a widely recognized technique for improving model performance. Regrettably, when training private ML models, many practitioners often overlook the privacy risks associated with hyperparameter optimization, which could potentially expose sensitive information about the underlying dataset. Currently, the sole existing approach to allow privacy-preserving hyperparameter optimization is to uniformly and randomly select hyperparameters for a number of runs, subsequently reporting the best-performing hyperparameter. In contrast, in non-private settings, practitioners commonly utilize "adaptive" hyperparameter optimization methods such as Gaussian process-based optimization, which select the next candidate based on information gathered from previous outputs. This substantial contrast between private and non-private hyperparameter optimization underscores a critical concern. In our paper, we introduce DP-HyPO, a pioneering framework for "adaptive" private hyperparameter optimization, aiming to bridge the gap between private and non-private hyperparameter optimization. To accomplish this, we provide a comprehensive differential privacy analysis of our framework. Furthermore, we empirically demonstrate the effectiveness of DP-HyPO on a diverse set of real-world datasets.

## 1 Introduction

In recent decades, modern deep learning has demonstrated remarkable advancements in various applications. Nonetheless, numerous training tasks involve the utilization of sensitive information pertaining to individuals, giving rise to substantial concerns regarding privacy [29, 6]. To address these concerns, the concept of differential privacy (DP) was introduced by [12, 13]. DP provides a mathematically rigorous framework for quantifying privacy leakage, and it has gained widespread

37th Conference on Neural Information Processing Systems (NeurIPS 2023).

acceptance as the most reliable approach for formally evaluating the privacy guarantees of machine learning algorithms.

When training deep learning models, the most popular method to ensure privacy is noisy (stochastic) gradient descent (DP-SGD) [4, 35]. DP-SGD typically resembles non-private gradient-based methods; however, it incorporates gradient clipping and noise injection. More specifically, each individual gradient is clipped to ensure a bounded $\ell_2$ norm. Gaussian noise is then added to the average gradient which is utilized to update the model parameters. These adjustments guarantee a bounded sensitivity of each update, thereby enforcing DP through the introduction of additional noise.

In both non-private and private settings, hyperparameter optimization (HPO) plays a crucial role in achieving optimal model performance. Commonly used methods for HPO include grid search (GS), random search (RS), and Bayesian optimization (BO). GS and RS approaches are typically non-adaptive, as they select the best hyperparameter from a predetermined or randomly selected set. While these methods are straightforward to implement, they can be computationally expensive and inefficient when dealing with large search spaces. As the dimensionality of hyperparameters increases, the number of potential trials may grow exponentially. To address this challenge, adaptive HPO methods such as Bayesian optimization have been introduced [34, 14, 40]. BO leverages a probabilistic model that maps hyperparameters to objective metrics, striking a balance between exploration and exploitation. BO quickly emerged as the default method for complex HPO tasks, offering improved efficiency and effectiveness compared to non-adaptive methods.

While HPO is a well-studied problem, the integration of a DP constraint into HPO has received little attention. Previous works on DP machine learning often neglect to account for the privacy cost associated with HPO [1, 39, 42, 42]. These works either assume that the best parameters are known in advance or rely on a supplementary public dataset that closely resembles the private dataset distribution, which is not feasible in most real-world scenarios.

Only recently have researchers turned to the important concept of honest HPO [28], where the privacy cost during HPO cannot be overlooked. Private HPO poses greater challenges compared to the non-private case for two primary reasons. First, learning with DP-SGD introduces additional hyperparameters (e.g., clipping norm, the noise scale, and stopping time), which hugely adds complexity to the search for optimal hyperparameters. Second, DP-SGD is more sensitive to the selection of hyperparameter combinations, with its performance largely influenced by this choice [28, 10, 31].

To tackle this challenging question, previous studies such as [24, 32] propose running the base algorithm with different hyperparameters a random number of times. They demonstrate that this approach significantly benefits privacy accounting, contrary to the traditional scaling of privacy guarantees with the square root of the number of runs (based on the composition properties from [19]). While these papers make valuable contributions, their approaches only allow for uniformly random subsampling from a finite and pre-fixed set of candidate hyperparameters at each run. As a result, any advanced technique from HPO literature that requires adaptivity is either prohibited or necessitates a considerable privacy cost (polynomially dependent on the number of runs), creating a substantial gap between non-private and private HPO methods.

Given these considerations, a natural question arises: *Can private hyperparameter optimization be adaptive, without a huge privacy cost?* In this paper, we provide an affirmative answer to this question.

## 1.1 Our Contributions

- **We introduce the pioneering adaptive private hyperparameter optimization framework, DP-HyPO,** which enables practitioners to adapt to previous runs and focus on potentially superior hyperparameters. DP-HyPO permits the flexible use of non-DP adaptive hyperparameter optimization methods, such as Gaussian process, for enhanced efficiency, while avoiding the substantial privacy costs due to composition. In contrast to the non-adaptive methods presented in [32, 24], our adaptive framework, DP-HyPO, effectively bridges the gap between private and non-private hyperparameter optimization. Importantly, our framework not only encompasses the aforementioned non-adaptive methods as special cases, but also seamlessly integrates virtually all conceivable adaptive methods into the framework.

- **We provide sharp DP guarantees for the adaptive private hyperparameter optimization.** Specifically, when the training procedure is executed multiple times, with each iteration being DP on its own, outputting the best repetition is DP ensured by the composition property. However, applying composition results in excessively loose privacy guarantees. Prior work in [24, 32] presents bounds that are either independent of the number of repetitions or depend logarithmically on it. Nevertheless, these results require that the hyperparameter selection for each iteration follows a uniform sampling distribution. In contrast, DP-HyPO allows arbitrary adaptive sampling distributions based on previous runs. Utilizing the Rényi DP framework, we offer a strict generalization of those uniform results by providing an accurate characterization of the Rényi divergence between the adaptive sampling distributions of neighboring datasets, without any stability assumptions.

- **Empirically, we observe that the Gaussian process-based DP-HyPO algorithm outperforms its uniform counterpart across several practical scenarios.** Generally, practitioners can integrate any non-private adaptive HPO methods into the DP-HyPO framework, opening up a vast range of adaptive private HPO algorithm possibilities. Furthermore, DP-HyPO grants practitioners the flexibility to determine the privacy budget allocation for adaptivity, empowering them to balance between the adaptivity and privacy loss when confronting various hyperparameter optimization challenges.

## 2 Preliminaries

### 2.1 Differential Privacy and Hyperparameter Optimization

Differential Privacy is a mathematically rigorous framework for quantifying privacy leakage. A DP algorithm promises that an adversary with perfect information about the entire private dataset in use – except for a single individual – would find it hard to distinguish between its presence or absence based on the output of the algorithm [12]. Formally, for $\varepsilon > 0$, and $0 \leq \delta < 1$, we consider a (randomized) algorithm $M : \mathcal{Z}^n \to \mathcal{Y}$ that takes as input a dataset.

**Definition 2.1** (Differential privacy). A randomized algorithm $M$ is $(\varepsilon, \delta)$-DP if for any neighboring dataset $D, D' \in \mathcal{Z}^n$ differing by an arbitrary sample, and for any event $E$, we have

$$\mathbb{P}[M(D) \in E] \leqslant \mathrm{e}^{\varepsilon} \cdot \mathbb{P}\left[M\left(D'\right) \in E\right] + \delta.$$

Here, $\varepsilon$ and $\delta$ are privacy parameters that characterize the privacy guarantee of algorithm $M$. One of the fundamental properties of DP is composition. When multiple DP algorithms are sequentially composed, the resulting algorithm remains private. The total privacy cost of the composition scales approximately with the square root of the number of compositions [19].

We now formalize the problem of hyperparameter optimization with DP guarantees, which builds upon the finite-candidate framework presented in [24, 32]. Specifically, we consider a set of base DP algorithms $M_\lambda : \mathcal{Z}^n \to \mathcal{Y}$, where $\lambda \in \Lambda$ represents a set of hyperparameters of interest, $\mathcal{Z}^n$ is the domain of datasets, and $\mathcal{Y}$ denotes the range of the algorithms. This set $\Lambda$ may be any infinite set, e.g., the cross product of the learning rate $\eta$ and clipping norm $R$ in DP-SGD. We require that the set $\Lambda$ is a measure space with an associated measure $\mu$. Common choices for $\mu$ include the counting measure or Lebesgue measure. We make a mild assumption that $\mu(\Lambda) < \infty$.

Based on the previous research [32], we make two simplifying assumptions. First, we assume that there is a total ordering on the range $\mathcal{Y}$, which allows us to compare two selected models based on their "performance measure", denoted by $q$. Second, we assume that, for hyperparameter optimization purposes, we output the trained model, the hyperparameter, and the performance measure. Specifically, for any input dataset $D$ and hyperparameter $\lambda$, the return value of $M_\lambda$ is $(x, q) \sim M_\lambda(D)$, where $x$ represents the combination of the model parameters and the hyperparameter $\lambda$, and $q$ is the (noisy) performance measure of the model.

### 2.2 Related Work

In this section, we focus on related work concerning private HPO, while deferring the discussion on non-private HPO to Appendix F.

Historically, research in DP machine learning has neglected the privacy cost associated with HPO [1, 39, 42]. It is only recently that researchers have begun to consider the honest HPO setting [28], in which the cost is taken into account.

A direct approach to addressing this issue involves composition-based analysis. If each training run of a hyperparameter satisfies DP, the entire HPO procedure also complies with DP through composition across all attempted hyperparameter values. However, the challenge with this method is that the privacy guarantee derived from accounting can be excessively loose, scaling polynomially with the number of runs.

Chaudhuri et al. [7] were the first to enhance the DP bounds for HPO by introducing additional stability assumptions on the learning algorithms. [24] made significant progress in enhancing DP bounds for HPO without relying on any stability properties of the learning algorithms. They proposed a simple procedure where a hyperparameter was randomly selected from a uniform distribution for each training run. This selection process was repeated a random number of times according to a geometric distribution, and the best model obtained from these runs was outputted. They showed that this procedure satisfied $(3\varepsilon, 0)$-DP as long as each training run of a hyperparameter was $(\varepsilon, 0)$-DP. Building upon this, [32] extended the procedure to accommodate negative binomial or Poisson distributions for the repeated uniform selection. They also offered more precise Rényi DP guarantees for this extended procedure. Furthermore, [8] explored a generalization of the procedure for top-$k$ selection, considering $(\varepsilon, \delta)$-DP guarantees.

In a related context, [28] explored a setting that appeared superficially similar to ours, as their title mentioned "adaptivity." However, their primary focus was on improving adaptive optimizers such as DP-Adam, which aimed to reduce the necessity of hyperparameter tuning, rather than the adaptive HPO discussed in this paper. Notably, in terms of privacy accounting, their approach only involved composing the privacy cost of each run without proposing any new method.

Another relevant area of research is DP selection, which encompasses well-known methods such as the exponential mechanism [25] and the sparse vector technique [13], along with subsequent studies. However, this line of research always assumes the existence of a low-sensitivity score function for each candidate, which is an unrealistic assumption for hyperparameter optimization.

## 3 DP-HyPO: General Framework for Private Hyperparameter Optimization

The obvious approach to the problem of differentially private hyperparameter optimization would be to run each base algorithm and simply return the best one. However, running such an algorithm on large hyperparameter space is not feasible due to the privacy cost growing linearly in the worst case.

While [24, 32] have successfully reduced the privacy cost for hyperparameter optimization from linear to constant, there are still two major drawbacks. First, none of the previous methods considers the case when the potential number of hyperparameter candidates is infinite, which is common in most hyperparameter optimization scenarios. In fact, we typically start with a range of hyperparameters that we are interested in, rather than a discrete set of candidates. Furthermore, prior methods are limited to the uniform sampling scheme over the hyperparameter domain $\Lambda$. In practice, this setting is unrealistic since we want to "adapt" the selection based on previous results. For instance, one could use Gaussian process to adaptively choose the next hyperparameter for evaluation, based on all the previous outputs. However, no adaptive hyperparameter optimization method has been proposed or analyzed under the DP constraint. In this paper, we bridge this gap by introducing the first DP adaptive hyperparameter optimization framework.

### 3.1 DP-HyPO Framework

To achieve adaptive hyperparameter optimization with differential privacy, we propose the DP-HyPO framework. Our approach keeps an adaptive sampling distribution $\pi$ at each iteration that reflects accumulated information.

Let $Q(D, \pi)$ be the procedure that randomly draws a hyperparameter $\lambda$ from the distribution[1] $\pi \in \mathcal{D}(\Lambda)$, and then returns the output from $M_\lambda(D)$. We allow the sampling distribution to depend on both the dataset and previous outputs, and we denote as $\pi^{(j)}$ the sampling distribution at the $j$-th

---

[1]Here, $\mathcal{D}(\Lambda)$ represents the space of probability densities on $\Lambda$.

iteration on dataset $D$. Similarly, the sampling distribution at the $j$-th iteration on the neighborhood dataset $D'$ is denoted as $\pi'^{(j)}$.

We now present the DP-HyPO framework, denoted as $\mathcal{A}(D, \pi^{(0)}, \mathcal{T}, C, c)$, in Framework 1. The algorithm takes a prior distribution $\pi^{(0)} \in \mathcal{D}(\Lambda)$ as input, which reflects arbitrary prior knowledge about the hyperparameter space. Another input is the distribution $\mathcal{T}$ of the total repetitions of training runs. Importantly, we require it to be a random variable rather than a fixed number to preserve privacy. The last two inputs are $C$ and $c$, which are upper and lower bounds of the density of any posterior sampling distributions. A finite $C$ and a positive $c$ are required to bound the privacy cost of the entire framework.

---

**Framework 1** DP-HyPO $\mathcal{A}(D, \pi^{(0)}, \mathcal{T}, C, c)$

---

Initialize $\pi^{(0)}$, a prior distribution over $\Lambda$.
Initialize the result set $A = \{\}$
Draw $T \sim \mathcal{T}$
**for** $j = 0$ to $T - 1$ **do**
    $(x, q) \sim Q(D, \pi^{(j)})$
    $A = A \cup \{(x, q)\}$
    Update $\pi^{(j+1)}$ based on $A$ according to any adaptive algorithm such that for all $\lambda \in \Lambda$,

$$c \le \frac{\pi^{(j+1)}(\lambda)}{\pi^{(0)}(\lambda)} \le C$$

**end for**
Output $(x, q)$ from $A$ with the highest $q$

---

Note that we intentionally leave the update rule for $\pi^{(j+1)}$ unspecified in Framework 1 to reflect the fact that any adaptive update rule that leverages information from previous runs can be used. However, for a non-private adaptive HPO update rule, the requirement of bounded adaptive density $c \le \frac{\pi^{(j+1)}(\lambda)}{\pi^{(0)}(\lambda)} \le C$ may be easily violated. In Section 3.2, We provide a simple projection technique to privatize any non-private update rules. In Section 4, we provide an instantiation of DP-HyPO using Gaussian process.

We now state our main privacy results for this framework in terms of Rényi Differential Privacy (RDP) [27]. RDP is a privacy measure that is more general than the commonly used $(\varepsilon, \delta)$-DP and provides tighter privacy bounds for composition. We defer its exact definition to Definition A.2 in the appendix.

We note that different distributions of the number of selections (iterations), $\mathcal{T}$, result in very different privacy guarantees. Here, we showcase the key idea for deriving the privacy guarantee of DP-HyPO framework by considering a special case when $\mathcal{T}$ follows a truncated negative binomial distribution[2] NegBin$(\theta, \gamma)$ (the same assumption as in [32]). In fact, as we show in the proof of Theorem 1 in Appendix A, the privacy bounds only depend on $\mathcal{T}$ directly through its probability generating function, and therefore one can adapt the proof to obtain the corresponding privacy guarantees for other probability families, for example, the Possion distribution considered in [32]. From here and on, unless otherwise specified, we will stick with $\mathcal{T} = $ NegBin$(\theta, \gamma)$ for simplicity. We also assume for simplicity that the prior distribution $\pi^{(0)}$ is a uniform distribution over $\Lambda$. We provide more detailed discussion of handling informed prior other than uniform distribution in Appendix D.

**Theorem 1.** *Suppose that $T$ follows truncated negative Binomial distribution $T \sim$ NegBin$(\theta, \gamma)$. Let $\theta \in (-1, \infty)$, $\gamma \in (0, 1)$, and $0 < c \le C$. Suppose for all $M_\lambda : \mathcal{Z}^n \to \mathcal{Y}$ over $\lambda \in \Lambda$, the base algorithms satisfy $(\alpha, \varepsilon)$-RDP and $(\hat{\alpha}, \hat{\varepsilon})$-RDP for some $\varepsilon, \hat{\varepsilon} \ge 0, \alpha \in (1, \infty)$, and $\hat{\alpha} \in [1, \infty)$. Then the DP-HyPO algorithm $\mathcal{A}(D, \pi^{(0)}, $ NegBin$(\theta, \gamma), C, c)$ satisfies $(\alpha, \varepsilon')$-RDP where*

$$\varepsilon' = \varepsilon + (1 + \theta) \cdot \left(1 - \frac{1}{\hat{\alpha}}\right) \hat{\varepsilon} + \left(\frac{\alpha}{\alpha - 1} + 1 + \theta\right) \log \frac{C}{c} + \frac{(1 + \theta) \cdot \log(1/\gamma)}{\hat{\alpha}} + \frac{\log \mathbb{E}[T]}{\alpha - 1}.$$

---

[2]Truncated negative binomial distribution is a direct generalization of the geometric distribution. See Appendix B for its definition.

To prove Theorem 1, one of our main technical contributions is Lemma A.4, which quantifies the Rényi divergence of the sampling distribution at each iteration between the neighboring datasets. We then leverage this crucial result and the probability generating function of $\mathcal{T}$ to bound the Rényi divergence in the output of $\mathcal{A}$. We defer the detailed proof to Appendix A.

Next, we present the case with pure DP guarantees. Recall the fact that $(\varepsilon, 0)$-DP is equivalent to $(\infty, \varepsilon)$-RDP [27]. When both $\alpha$ and $\hat{\alpha}$ tend towards infinity, we easily obtain the following theorem in terms of $(\varepsilon, 0)$-DP.

**Theorem 2.** *Suppose that $T$ follows truncated negative Binomial distribution $T \sim \text{NegBin}(\theta, \gamma)$. Let $\theta \in (-1, \infty)$ and $\gamma \in (0, 1)$. If all the base algorithms $M_\lambda$ satisfies $(\varepsilon, 0)$-DP, then the DP-HyPO algorithm $\mathcal{A}(D, \pi^{(0)}, \text{NegBin}(\theta, \gamma), C, c)$ satisfies $\left( (2 + \theta)\left(\varepsilon + \log \frac{C}{c}\right), 0 \right)$-DP.*

Theorem 1 and Theorem 2 provide practitioners the freedom to trade off between allocating more DP budget to enhance the base algorithm or to improve adaptivity. In particular, a higher value of $\frac{C}{c}$ signifies greater adaptivity, while a larger $\varepsilon$ improves the performance of base algorithms.

### 3.1.1 Uniform Optimization Method as a Special Case

We present the uniform hyperparameter optimization method [32, 23] in Algorithm 2, which is a special case of our general DP-HyPO Framework with $C = c = 1$. Essentially, this algorithm never updates the sampling distribution $\pi$.

---

**Algorithm 2** Uniform Hyperparameter Optimization $\mathcal{U}(D, \theta, \gamma, \Lambda)$

---
    Let $\pi = \text{Unif}(\{1, ..., |\Lambda|\})$, and $A = \{\}$
    Draw $T \sim \text{NegBin}(\theta, \gamma)$
    **for** $j = 0$ to $T - 1$ **do**
        $(x, q) \sim Q(D, \pi)$
        $A = A \cup \{(x, q)\}$
    **end for**
    Output $(x, q)$ from $A$ with the highest $q$

---

Our results in Theorem 1 and Theorem 2 generalize the main technical results of [32, 24]. Specifically, when $C = c = 1$ and $\Lambda$ is a finite discrete set, our Theorem 1 precisely recovers Theorem 2 in [32]. Furthermore, when we set $\theta = 1$, the truncated negative binomial distribution reduces to the geometric distribution, and our Theorem 2 recovers Theorem 3.2 in [24] .

### 3.2 Practical Recipe to Privatize HPO Algorithms

In the DP-HyPO framework, we begin with a prior and adaptively update it based on the accumulated information. However, for privacy purposes, we require the density $\pi^{(j)}$ to be bounded by some constants $c$ and $C$, which is due to the potential privacy leakage when updating $\pi^{(j)}$ based on the history. It is crucial to note that this distribution $\pi^{(j)}$ can be significantly different from the distribution $\pi'^{(j)}$ if we were given a different input dataset $D'$. Therefore, we require the probability mass/density function to satisfy $\frac{c}{\mu(\Lambda)} \leq \pi^{(j)}(\lambda) \leq \frac{C}{\mu(\Lambda)}$ for all $\lambda \in \Lambda$ to control the privacy loss due to adaptivity.

This requirement is not automatically satisfied and typically necessitates modifications to current non-private HPO methods. To address this challenge, we propose a general recipe to modify any non-private method. The idea is quite straightforward: throughout the algorithm, we maintain a non-private version of the distribution density $\pi^{(j)}$. When sampling from the space $\Lambda$, we perform a projection from $\pi^{(j)}$ to the space consisting of bounded densities. Specifically, we define the space of essentially bounded density functions by $S_{C,c} = \{f \in \Lambda^{\mathbb{R}^+} : \text{ess sup } f \leq \frac{C}{\mu(\Lambda)}, \text{ess inf } f \geq \frac{c}{\mu(\Lambda)}, \int_{\alpha \in \Lambda} f(\alpha) d\alpha = 1\}$. For such a space to be non-empty, we require that $c \leq 1 \leq C$, where $\mu$ is the measure on $\Lambda$. This condition is well-defined as we assume $\mu(\Lambda) < \infty$.

To privatize $\pi^{(j)}$ at the $j$-th iteration, we project it into the space $S_{C,c}$, by solving the following convex functional programming problem:

$$\min_{f} \ \|f - \pi^{(j)}\|_2,$$
$$\text{s.t. } f \in S_{C,c}. \tag{3.1}$$

Note that this is a convex program since $S_{C,c}$ is convex and closed. We denote the output from this optimization problem by $\mathcal{P}_{S_{C,c}}(\pi^{(j)})$. Theoretically, problem (3.1) allows the hyperparameter space $\Lambda$ to be general measurable space with arbitrary topological structure. However, empirically, practitioners need to discretize $\Lambda$ to some extent to make the convex optimization computationally feasible. Compared to the previous work, our formulation provides the most general characterization of the problem and allows pratitioners to *adaptively* and *iteratively* choose a proper discretization as needed. Framework 1 tolerates a much finer level of discretization than the previous method, as the performance of latter degrades fast when the number of candidates increases. We also provide examples using CVX to solve this problem in Section 4.2. In Appendix C, we discuss about its practical implementation, and the connection to information projection.

# 4 Application: DP-HyPO with Gaussian Process

In this section, we provide an instantiation of DP-HyPO using Gaussian process (GP) [38]. GPs are popular non-parametric Bayesian models frequently employed for hyperparameter optimization. At the meta-level, GPs are trained to generate surrogate models by establishing a probability distribution over the performance measure $q$. While traditional GP implementations are not private, we leverage the approach introduced in Section 3.2 to design a private version that adheres to the bounded density contraint.

We provide the algorithmic description in Section 4.1 and the empircal evaluation in Section 4.2.

## 4.1 Algorithm Description

The following Algorithm ($\mathcal{AGP}$) is a private version of Gaussian process for hyperparameter tuning. In Algorithm 3, we utilize GP to construct a surrogate model that generates probability distributions

---

**Algorithm 3** DP-HyPO with Gaussian process $\mathcal{AGP}(D, \theta, \gamma, \tau, \beta, \Lambda, C, c)$

---

Initialize $\pi^{(0)} = \text{Unif}(\Lambda)$, and $A = \{\}$
Draw $T \sim \text{NegBin}(\theta, \gamma)$
**for** $t = 0$ to $T - 1$ **do**
    Truncate the density of current $\pi^{(t)}$ to be bounded into the range of $[c, C]$ by projecting to $S_{C,c}$.
$$\tilde{\pi}^{(t)} = \mathcal{P}_{S_{C,c}}(\pi^{(t)}).$$
    Sample $(x, q) \sim Q(D, \tilde{\pi}^{(j)})$, and update $A = A \cup \{(x, q)\}$
    Update mean estimation and variance estimation of the Gaussian process $\mu_\lambda, \sigma_\lambda^2$, and get the score as $s_\lambda = \mu_\lambda + \tau \sigma_\lambda$.
    Update true (untruncated) posterior $\pi^{(t+1)}$ with softmax, by $\pi^{(t+1)}(\lambda) = \frac{\exp(\beta \cdot s_\lambda)}{\int_{\lambda' \in \Lambda} \exp(\beta \cdot s'_\lambda)}$.
**end for**
Output $(x, q)$ from $A$ with the highest $q$

---

for the performance measure $q$. By estimating the mean and variance, we assign a "score" to each hyperparameter $\lambda$, known as the estimated upper confidence bound (UCB). The weight factor $\tau$ controls the balance between exploration and exploitation, where larger weights prioritize exploration by assigning higher scores to hyperparameters with greater uncertainty.

To transform these scores into a sampling distribution, we apply the softmax function across all hyperparameters, incorporating the parameter $\beta$ as the inverse temperature. A higher value of $\beta$ signifies increased confidence in the learned scores for each hyperparameter.

## 4.2 Empirical Evaluations

We now evaluate the performance of our GP-based DP-HyPO (referred to as "GP") in various settings. Since DP-HyPO is the first adaptive private hyperparameter optimization method of its kind, we compare it to the special case of Uniform DP-HyPO (Algorithm 2), referred to as "Uniform", as proposed in [24, 32]. In this demonstration, we consider two pragmatic privacy configurations: the white-box setting and the black-box setting, contingent on whether adaptive HPO algorithms incur extra privacy cost. In the white-box scenario (Section 4.2.1 and 4.2.2), we conduct experiments involving training deep learning models on both the MNIST dataset and CIFAR-10 dataset. Conversely, when considering the black-box setting (Section 4.2.3), our attention shifts to a real-world Federated Learning (FL) task from the industry. These scenarios provide meaningful insights into the effectiveness and applicability of our GP-based DP-HyPO approach.

### 4.2.1 MNIST Simulation

We begin with the white-box scenario, in which the data curator aims to provide overall protection to the published model. In this context, to accommodate adaptive HPO algorithms, it becomes necessary to reduce the budget allocated to the base algorithm.

In this section, we consider the MNIST dataset, where we employ DP-SGD to train a standard CNN. The base algorithms in this case are different DP-SGD models with varying hyperparameters, and we evaluate each base algorithm based on its accuracy. Our objective is to identify the best hyperparameters that produce the most optimal model within a given total privacy budget.

Specifically, we consider two variable hyperparameters: the learning rate $\eta$ and clipping norm $R$, while keeping the other parameters fixed. We ensure that both the GP algorithm and the Uniform algorithm operate under the same total privacy budget, guaranteeing a fair comparison.

Due to constraints on computational resources, we conduct a semi-real simulation using the MNIST dataset. For both base algorithms (with different noise multipliers), we cache the mean accuracy of 5 independently trained models for each discretized hyperparameter and treat that as a proxy for the "actual accuracy" of the hyperparameter. Each time we sample the accuracy of a hyperparameter, we add a Gaussian noise with a standard deviation of $0.1$ to the cached mean. We evaluate the performance of the output model based on the "actual accuracy" corresponding to the selected hyperparameter. Further details on the simulation and parameter configuration can be found in Appendix E.1.

In the left panel of Figure 1, we demonstrated the comparison of performance of the Uniform and GP methods with total privacy budget $\varepsilon = 15^3$ and $\delta = 1e - 5$. The accuracy reported is the actual accuracy of the output hyperparameter. From the figure, we see that when $T$ is very small ($T < 8$), GP method is slightly worse than Uniform method as GP spends $\log(C/c)$ budget less than Uniform method for each base algorithm (the cost of adaptivity). However, we see that after a short period of exploration, GP consistently outperform Uniform, mostly due to the power of being adaptive. The superiority of GP is further demonstrated in Table 1, aggregating over geometric distribution.

### 4.2.2 CIFAR-10 Simulation

When examining the results from MNIST, a legitimate critique arises: our DP-Hypo exhibits only marginal superiority over its uniform counterpart, which questions the assertion that adaptivity holds significant value. Our conjecture is that the hyperparameter landscape of MNIST is relatively uncomplicated, which limits the potential benefits of adaptive algorithms.

To test the hypothesis, we conduct experiments on the CIFAR-10 dataset, with a setup closely mirroring the previous experiment: we employ the same CNN model for training, and optimize the same set of hyperparameters, which are the learning rate $\eta$ and clipping norm $R$. The primary difference lies in how we generate the hyperparameter landscape. Given that a single run on CIFAR-10 is considerably more time-consuming than on MNIST, conducting multiple runs for every hyperparameter combination is unfeasible. To address this challenge, we leverage BoTorch [3],

---

[3]The $\varepsilon$ values are seemingly very large. Nonetheless, the reported privacy budget encompasses the overall cost of the entire HPO, which is typically overlooked in the existing literature. Given that HPO roughly incurs three times the privacy cost of the base algorithm, an $\varepsilon$ as high as 15 could be reported as only 5 in many other works.

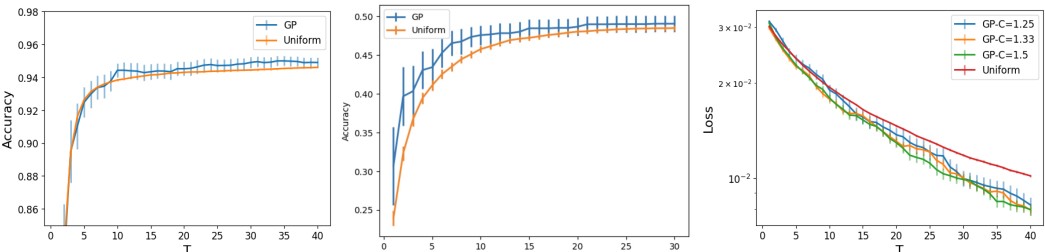

Figure 1: Left: The accuracy of the output hyperparameter in MNIST semi-real simulation, with $\varepsilon = 15$, $\delta = 0.00001$. Middle: The accuracy of the output hyperparameter in CIFAR-10, with $\varepsilon = 12$, $\delta = 0.00001$. Right: The loss of the output hyperparameter in FL. Error bars stands for $95\%$ confidence. Curves for GP are calculated by averaging $400$ independent runs, and curves for Uniform are calculated by averaging 10000 independent runs. For a clearer demonstration, we compare the performance for each fixed value of $T$, and recognize that the actual performance is a weighted average across different values of $T$.

an open-sourced library for HPO, to generate the landscape. Since we operate in the white-box setting, where the base algorithms have distinct privacy budgets for the uniform and adaptive scenarios, we execute 50 runs and generate the landscape for each case, including the mean ($\mu_\lambda$) and standard error ($\sigma_\lambda$) of accuracy for each hyperparameter combination $\lambda$. When the algorithm (GP or Uniform) visits a specific $\lambda$, our oracle returns a noisy score $q(\lambda)$ drawn from a normal distribution of $N(\mu_\lambda, \sigma_\lambda)$. A more detailed description of our landscapes and parameter configuration can be found in Appendix E.2.

In the middle of Figure 1, we showcase a performance comparison between the Uniform and GP methods with a total privacy budget of $\varepsilon = 12$ and $\delta = 1e - 5$. Clearly, GP consistently outperforms the Uniform method, with the largest performance gap occurring when the number of runs is around 10.

### 4.2.3 Federated Learning

In this section, we move to the black-box setting, where the privacy budget allocated to the base algorithm remains fixed, while we allow extra privacy budget for HPO. That being said, the adaptivity can be achieved without compromising the utility of the base algorithm.

We explore another real-world scenario: a Federated Learning (FL) task conducted on a proprietary dataset [4] from industry. Our aim is to determine the optimal learning rates for the central server (using AdaGrad) and the individual users (using SGD). To simulate this scenario, we once again rely on the landscape generated by BoTorch [3], as shown in Figure 3 in Appendix E.3.

Under the assumption that base algorithms are black-box models with fixed privacy costs, we proceed with HPO while varying the degree of adaptivity. The experiment results are visualized in the right panel of Figure 1, and Table 2 presents the aggregated performance data.

We consistently observe that GP outperforms Uniform in the black-box setting. Furthermore, our findings suggest that allocating a larger privacy budget to the GP method facilitates the acquisition of adaptive information, resulting in improved performance in HPO. This highlights the flexibility of GP in utilizing privacy resources effectively.

| Geometric($\gamma$) | 0.001 | 0.002 | 0.003 | 0.005 | 0.01 | 0.02 | 0.025 | 0.03 |
|---|---|---|---|---|---|---|---|---|
| GP | 0.946 | 0.948 | 0.948 | 0.947 | 0.943 | 0.937 | 0.934 | 0.932 |
| Uniform | 0.943 | 0.945 | 0.945 | 0.944 | 0.940 | 0.935 | 0.932 | 0.929 |

Table 1: Accuracy of MNIST using Geometric Distribution with various different values of $\gamma$ for Uniform and GP methods. Each number is the mean of 200 runs.

---

[4]We have to respect confidentiality constraints that limit our ability to provide extensive details about this dataset.

| Geometric($\gamma$) | 0.001 | 0.002 | 0.003 | 0.005 | 0.01 | 0.02 | 0.025 | 0.03 |
|---|---|---|---|---|---|---|---|---|
| GP (C = 1.25) | 0.00853 | 0.0088 | 0.00906 | 0.00958 | 0.0108 | 0.0129 | 0.0138 | 0.0146 |
| GP (C = 1.33) | 0.00821 | 0.00847 | 0.00872 | 0.00921 | 0.0104 | 0.0123 | 0.0132 | 0.0140 |
| GP (C = 1.5) | 0.00822 | 0.00848 | 0.00872 | 0.00920 | 0.0103 | 0.0123 | 0.0131 | 0.0130 |
| Uniform | 0.0104 | 0.0106 | 0.0109 | 0.0113 | 0.0123 | 0.0141 | 0.0149 | 0.0156 |

Table 2: Loss of FL using Geometric Distribution with various different values of $\gamma$ for Uniform and GP methods with different choice of $C$ and $c = 1/C$. Each number is the mean of 200 runs.

## 5    Conclusion

In conclusion, this paper presents a novel framework, DP-HyPO. As the first adaptive HPO framework with sharp DP guarantees, DP-HyPO effectively bridges the gap between private and non-private HPO. Our work encompasses the random search method by [24, 32] as a special case, while also granting practitioners the ability to adaptively learn better sampling distributions based on previous runs. Importantly, DP-HyPO enables the conversion of any non-private adaptive HPO algorithm into a private one. Our framework proves to be a powerful tool for professionals seeking optimal model performance and robust DP guarantees.

The DP-HyPO framework presents two interesting future directions. One prospect involves an alternative HPO specification which is practically more favorable. Considering the extensive literature on HPO, there is a significant potential to improve the empirical performance by leveraging more advanced HPO methods. Secondly, there is an interest in establishing a theoretical utility guarantee for DP-HyPO. By leveraging similar proof methodologies to those in Theorem 3.3 in [24], it is feasible to provide basic utility guarantees for the general DP-HyPO, or for some specific configurations within DP-HyPO.

## 6    Acknowledgements

The authors would like to thank Max Balandat for his thoughtful comments and insights that helped us improve the paper.

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

# A   Proofs of the technical results

## A.1   Proof of Main Results

First, we define Rényi divergence as follows.

**Definition A.1** (Rényi Divergences). Let $P$ and $Q$ be probability distributions on a common space $\Omega$. Assume that $P$ is absolutely continuous with respect to $Q$ - i.e., for all measurable $E \subset \Omega$, if $Q(E) = 0$, then $P(E) = 0$. Let $P(x)$ and $Q(x)$ denote the densities of $P$ and $Q$ respectively. The KL divergence from $P$ to $Q$ is defined as

$$D_1(P\|Q) := \underset{X \leftarrow P}{\mathbb{E}} \left[ \log \left( \frac{P(X)}{Q(X)} \right) \right] = \int_\Omega P(x) \log \left( \frac{P(x)}{Q(x)} \right) dx.$$

The max divergence from $P$ to $Q$ is defined as

$$D_\infty(P\|Q) := \sup \left\{ \log \left( \frac{P(E)}{Q(E)} \right) : P(E) > 0 \right\}.$$

For $\alpha \in (1, \infty)$, the Rényi divergence from $P$ to $Q$ of order $\alpha$ is defined as

$$\begin{aligned}
D_\alpha(P\|Q) &:= \frac{1}{\alpha - 1} \log \left( \underset{X \leftarrow P}{\mathbb{E}} \left[ \left( \frac{P(X)}{Q(X)} \right)^{\alpha - 1} \right] \right) \\
&= \frac{1}{\alpha - 1} \log \left( \underset{X \leftarrow Q}{\mathbb{E}} \left[ \left( \frac{P(X)}{Q(X)} \right)^{\alpha} \right] \right) \\
&= \frac{1}{\alpha - 1} \log \left( \int_Q P(x)^\alpha Q(x)^{1-\alpha} dx \right).
\end{aligned}$$

We now present the definition of Rényi DP (RDP) in [27].

**Definition A.2** (Rényi Differential Privacy). A randomized algorithm $M : \mathcal{X}^n \to \mathcal{Y}$ is $(\alpha, \varepsilon)$-Rényi differentially private if, for all neighbouring pairs of inputs $D, D' \in \mathcal{X}^n$, $D_\alpha(M(x)\|M(x')) \le \varepsilon$.

We define some additional notations for the sake of the proofs. In algorithm 1, for any $1 \le j \le T$, and neighboring dataset $D$ and $D'$, we define the following notations for any $y = (x, q) \in \mathcal{Y}$, the totally ordered range set.

$$\begin{aligned}
P_j(y) = \mathbb{P}_{\tilde{y} \sim Q(D, \pi^{(j)})}(\tilde{y} = y) \quad &\text{and} \quad P'_j(y) = \mathbb{P}_{\tilde{y} \sim Q(D', \pi'^{(j)})}(\tilde{y} = y) \\
P_j(\le y) = \mathbb{P}_{\tilde{y} \sim Q(D, \pi^{(j)})}(\tilde{y} \le y) \quad &\text{and} \quad P'_j(\le y) = \mathbb{P}_{\tilde{y} \sim Q(D', \pi'^{(j)})}(\tilde{y} \le y) \\
P_j(< y) = \mathbb{P}_{\tilde{y} \sim Q(D, \pi^{(j)})}(\tilde{y} < y) \quad &\text{and} \quad P'_j(< y) = \mathbb{P}_{\tilde{y} \sim Q(D', \pi'^{(j)})}(\tilde{y} < y).
\end{aligned}$$

By these definitions, we have $P_j(\le y) = P_j(< y) + P_j(y)$, and $P'_j(\le y) = P'_j(< y) + P'_j(y)$. And additionally, we have

$$\begin{aligned}
\frac{P_j(y)}{P'_j(y)} = \frac{\int_{\lambda \in \Lambda} \mathbb{P}(M_\lambda(D) = y) \pi^{(j)}(\lambda) d\lambda}{\int_{\lambda \in \Lambda} \mathbb{P}(M_\lambda(D') = y) \pi'^{(j)}(\lambda) d\lambda} &\le \sup_{\lambda \in \Lambda} \frac{\mathbb{P}(M_\lambda(D) = y) \pi^{(j)}(\lambda)}{\mathbb{P}(M_\lambda(D') = y) \pi'^{(j)}(\lambda)} \\
&\le \frac{C}{c} \cdot \sup_{\lambda \in \Lambda} \frac{\mathbb{P}(M_\lambda(D) = y)}{\mathbb{P}(M_\lambda(D') = y)}.
\end{aligned} \tag{A.1}$$

Here, the first inequality follows from the simple property of integration, and the second inequality follows from the fact that $\pi^{(j)}$ has bounded density between $c$ and $C$. Similarly, we have

$$\frac{P_j(\le y)}{P'_j(\le y)} \le \frac{C}{c} \cdot \sup_{\lambda \in \Lambda} \frac{\mathbb{P}(M_\lambda(D) \le y)}{\mathbb{P}(M_\lambda(D') \le y)}, \tag{A.2}$$

and

$$\frac{P_j(< y)}{P'_j(< y)} \le \frac{C}{c} \cdot \sup_{\lambda \in \Lambda} \frac{\mathbb{P}(M_\lambda(D) < y)}{\mathbb{P}(M_\lambda(D') < y)}. \tag{A.3}$$

Note that $D$ and $D'$ are neighboring datasets, and $M_\lambda$ satisfies some DP guarantees. So the ratio $\frac{\mathbb{P}(M_\lambda(D) \in E)}{\mathbb{P}(M_\lambda(D') \in E)}$ for any event $E$ can be bounded.

For simplicity, we define the inner product of a distribution $\pi$ with the vector $\boldsymbol{M}(D) = (\mathbb{P}(M_\lambda(D) = y) : \lambda \in \Lambda)$ as

$$\pi \cdot \boldsymbol{M}(D) := \int_{\lambda \in \Lambda} \mathbb{P}(M_\lambda(D) = y)\pi(\lambda)d\lambda. \tag{A.4}$$

Now, we define additional notations to bound the probabilities. Recall $S_{C,s}$ is given by $\{f \in \Lambda^{\mathbb{R}^+} :$ ess sup $f \le C$, ess inf $f \ge c$, $\int_{\alpha \in \Lambda} f(\alpha)d\alpha = 1.\}$. It is straightforward to see this is a compact set as it is the intersection of three compact sets. We define

$$P^+(y) := \sup_{\pi \in S_{C,c}} \int_{\lambda \in \Lambda} \mathbb{P}(M_\lambda(D) = y)\pi^{(j)}(\lambda)d\lambda = \pi^+ \cdot \boldsymbol{M}(D), \tag{A.5}$$

where $\pi^+$ is the distribution that achieves the supreme in the compact set $S_{C,c}$. Similarly, we define $P'^-(y)$ for $D'$ as given by

$$P'^-(y) := \inf_{\pi \in S_{C,c}} \int_{\lambda \in \Lambda} \mathbb{P}(M_\lambda(D') = y) \cdot \pi'^{(j)}(\lambda)d\lambda = \pi'^- \cdot \boldsymbol{M}. \tag{A.6}$$

Similarly, we can define $P'^+(y)$ and $P^-(y)$ accordingly. From the definition, we know that

$$P^-(y) \le P_j(y) \le P^+(y) \quad \text{and} \quad P'^-(y) \le P'_j(y) \le P'^+(y). \tag{A.7}$$

We also have

$$\frac{P^+(y)}{P'^-(y)} = \frac{\pi^* \cdot \boldsymbol{M}(D)}{\pi'^- \cdot \boldsymbol{M}(D')} \le \sup_\lambda \frac{\mathbb{P}(M_\lambda(D) = y)}{\mathbb{P}(M_\lambda(D') = y)} \cdot \frac{C}{c}. \tag{A.8}$$

It is similar to define

$$P^+(\le y) := \sup_{\pi \in S_{C,c}} \int_{\lambda \in \Lambda} \mathbb{P}(M_\lambda(D) \le y) \quad \text{and} \quad P'^+(\le y) := \sup_{\pi \in S_{C,c}} \int_{\lambda \in \Lambda} \mathbb{P}(M_\lambda(D') \le y)$$

$$P^-(\le y) := \inf_{\pi \in S_{C,c}} \int_{\lambda \in \Lambda} \mathbb{P}(M_\lambda(D) \le y) \quad \text{and} \quad P'^-(\le y) := \inf_{\pi \in S_{C,c}} \int_{\lambda \in \Lambda} \mathbb{P}(M_\lambda(D') \le y)$$

$$P^+(< y) := \sup_{\pi \in S_{C,c}} \int_{\lambda \in \Lambda} \mathbb{P}(M_\lambda(D) < y) \quad \text{and} \quad P'^+(< y) := \sup_{\pi \in S_{C,c}} \int_{\lambda \in \Lambda} \mathbb{P}(M_\lambda(D') < y)$$

$$P^-(< y) := \inf_{\pi \in S_{C,c}} \int_{\lambda \in \Lambda} \mathbb{P}(M_\lambda(D) < y) \quad \text{and} \quad P'^-(< y) := \inf_{\pi \in S_{C,c}} \int_{\lambda \in \Lambda} \mathbb{P}(M_\lambda(D') < y).$$

Following the exact same proof, we have

$$P^-(\le y) \le P_j(\le y) \le P^+(\le y) \quad \text{and} \quad P'^-(\le y) \le P'_j(\le y) \le P'^+(\le y) \tag{A.9}$$

$$P^-(< y) \le P_j(< y) \le P^+(< y) \quad \text{and} \quad P'^-(< y) \le P'_j(< y) \le P'^+(< y) \tag{A.10}$$

$$\frac{P^+(\le y)}{P'^-(\le y)} \le \sup_\lambda \frac{\mathbb{P}(M_\lambda(D) \le y)}{\mathbb{P}(M_\lambda(D') \le y)} \cdot \frac{C}{c} \quad \text{and} \quad \frac{P^+(< y)}{P'^-(< y)} \le \sup_\lambda \frac{\mathbb{P}(M_\lambda(D) < y)}{\mathbb{P}(M_\lambda(D') < y)} \cdot \frac{C}{c}. \tag{A.11}$$

It is also straightforward to verify from the definition that

$$P^+(\le y) = P^+(< y) + P^+(y) \quad \text{and} \quad P'^+(\le y) = P'^+(< y) + P'^+(y) \tag{A.12}$$

$$P^+ - (\le y) = P^-(< y) + P^-(y) \quad \text{and} \quad P'^-(\le y) = P'^-(< y) + P'^-(y). \tag{A.13}$$

**Lemma A.3.** *Suppose if $a_\lambda, b_\lambda$ are non-negative and $c_\lambda, c'_\lambda$ are positive for all $\lambda$. Then we have*

$$\frac{\sum_\lambda a_\lambda c_\lambda}{\sum_\lambda b_\lambda c'_\lambda} \le \frac{\sum_\lambda a_\lambda}{\sum_\lambda b_\lambda} \cdot \sup_{\lambda, \lambda'} \left| \frac{c_\lambda}{c'_\lambda} \right|.$$

*Proof of Lemma A.3.* This lemma is pretty straight forward by comparing the coefficient for each term in the full expansion. Specifically, we re-write the inequality as

$$\sum_\lambda a_\lambda c_\lambda \sum_{\lambda'} b'_\lambda \le \sum_\lambda a_\lambda \sum_{\lambda'} b'_\lambda c'_\lambda \cdot \sup_{\lambda, \lambda'} \left| \frac{c_\lambda}{c'_\lambda} \right|. \tag{A.14}$$

For each term $a_\lambda b'_\lambda$, its coefficient on the left hand side of (A.14) is $c_\lambda$, but its coefficient on the right hand side of (A.14) is $c'_\lambda \cdot \sup_{\lambda,\lambda'} \left| \frac{c_\lambda}{c'_\lambda} \right|$. Since we always have $c'_\lambda \cdot \sup_{\lambda,\lambda'} \left| \frac{c_\lambda}{c'_\lambda} \right| \geq c_\lambda$, and $a_\lambda b'_\lambda \geq 0$, we know the inequality (A.14) holds. $\qquad\square$

Next, in order to present our results in terms of RDP guarantees, we prove the following lemma.

**Lemma A.4.** *The Rényi divergence between $P^+$ and $P^-$ is be bounded as follows:*

$$\mathrm{D}_\alpha(P^+\|P'^-) \leq \frac{\alpha}{\alpha-1} \log \frac{C}{c} + \sup_{\lambda\in\Lambda} \mathrm{D}_\alpha\left(M_\lambda(D)\|M_\lambda(D')\right)$$

*Proof of Lemma A.4.* We write that

$$e^{(\alpha-1)\mathrm{D}_\alpha(P^+\|P'^-)} = \sum_{y\in\mathcal{Y}} P^+(y)^\alpha \cdot P'^-(y)^{1-\alpha} = \sum_{y\in\mathcal{Y}} \frac{\left(\sum_\lambda \pi^+(\lambda)\mathbb{P}(M_\lambda(D)=y)\right)^\alpha}{\left(\sum_\lambda \pi'^-(\lambda)\mathbb{P}(M_\lambda(D')=y)\right)^{\alpha-1}} \quad \text{(A.15)}$$

Here, $\pi^+$ and $\pi'^-$ are defined in (A.5) and (A.6), so they are essentially $\pi_y^+$ and $\pi_y'^-$ as they depend on the value of $y$. Therefore, we need to "remove" this dependence on $y$ to leverage the RDP guarantees for each base algorithm $M_\lambda$. We accomplish this task by bridging via $\pi$, the uniform density on $\Lambda$ (that is $\pi(\lambda) = \pi(\lambda')$ for any $\lambda, \lambda' \in \Lambda$). Specifically, we define $a_\lambda = \pi(\lambda)\mathbb{P}(M_\lambda(D) = y)$, $b_\lambda = \pi(\lambda)\mathbb{P}(M_\lambda(D') = y)$, $c_\lambda = \frac{\pi_y^+(\lambda)}{\pi(\lambda)}$, and $c'_\lambda = \frac{\pi_y'^-(\lambda)}{\pi(\lambda)}$. We see that

$$\sup_{\lambda,\lambda'} \left| \frac{c_\lambda}{c'_\lambda} \right| = \sup_{\lambda,\lambda'} \left| \frac{\pi_y^+(\lambda)/\pi(\lambda)}{\pi_y'^-(\lambda')/\pi(\lambda')} \right| = \sup_{\lambda,\lambda'} \left| \frac{\pi_y^+(\lambda))}{\pi_y'^-(\lambda')} \right| \leq C/c, \quad \text{(A.16)}$$

since $\pi$ is the uniform, and $\pi_y^+$ and $\pi_y'^-$ belongs to $S_{C,c}$. We now apply Lemma A.3 with the above notations for each $y$ to (A.15), and we have

$$\sum_{y\in\mathcal{Y}} \frac{\left(\sum_\lambda \pi^+(\lambda)\mathbb{P}(M_\lambda(D)=y)\right)^\alpha}{\left(\sum_\lambda \pi'^-(\lambda)\mathbb{P}(M_\lambda(D')=y)\right)^{\alpha-1}}$$

$$= \sum_{y\in\mathcal{Y}} \frac{\left(\sum_\lambda \pi(\lambda)\mathbb{P}(M_\lambda(D)=y)\cdot\frac{\pi^+(\lambda)}{\pi(\lambda)}\right)^{\alpha-1}\left(\sum_\lambda \pi(\lambda)\mathbb{P}(M_\lambda(D)=y)\cdot\frac{\pi^+(\lambda)}{\pi(\lambda)}\right)}{\left(\sum_\lambda \pi(\lambda)\mathbb{P}(M_\lambda(D')=y)\cdot\frac{\pi'^-(\lambda)}{\pi(\lambda)}\right)^{\alpha-1}}$$

$$= \sum_{y\in\mathcal{Y}} \frac{\left(\sum_\lambda a_\lambda \cdot c_\lambda\right)^{\alpha-1}\left(\sum_\lambda \pi(\lambda)\mathbb{P}(M_\lambda(D)=y)\cdot\frac{\pi^+(\lambda)}{\pi(\lambda)}\right)}{\left(\sum_\lambda b_\lambda \cdot c'_\lambda\right)^{\alpha-1}}$$

$$\leq \sum_{y\in\mathcal{Y}} \sup_{\lambda,\lambda'} \left|\frac{c_\lambda}{c'_\lambda}\right|^{\alpha-1} \frac{\left(\sum_\lambda a_\lambda\right)^{\alpha-1}\left(\sum_\lambda \pi(\lambda)\mathbb{P}(M_\lambda(D)=y)\cdot\frac{\pi^+(\lambda)}{\pi(\lambda)}\right)}{\left(\sum_\lambda b_\lambda\right)^{\alpha-1}}$$

$$= \sum_{y\in\mathcal{Y}} \sup_{\lambda,\lambda'} \left|\frac{c_\lambda}{c'_\lambda}\right|^{\alpha-1} \frac{\left(\sum_\lambda a_\lambda\right)^{\alpha-1}\left(\sum_\lambda a_\lambda \cdot c_\lambda\right)}{\left(\sum_\lambda b_\lambda\right)^{\alpha-1}}$$

$$\leq \sum_{y\in\mathcal{Y}} \sup_{\lambda,\lambda'} \left|\frac{c_\lambda}{c'_\lambda}\right|^{\alpha-1} \frac{\left(\sum_\lambda a_\lambda\right)^{\alpha-1}\left(\sum_\lambda a_\lambda\right)\cdot\sup_\lambda c_\lambda}{\left(\sum_\lambda b_\lambda\right)^{\alpha-1}}$$

$$\leq \sum_{y\in\mathcal{Y}} \left(\frac{C}{c}\right)^{\alpha-1} \frac{\left(\sum_\lambda a_\lambda\right)^{\alpha-1}\left(\sum_\lambda a_\lambda\right)\cdot\left(\frac{C}{c}\right)}{\left(\sum_\lambda b_\lambda\right)^{\alpha-1}}$$

$$= \sum_{y\in\mathcal{Y}} \left(\frac{C}{c}\right)^\alpha \cdot \frac{\left(\sum_\lambda \pi(\lambda)\mathbb{P}(M_\lambda(D)=y)\right)^\alpha}{\left(\sum_\lambda \pi(\lambda)\mathbb{P}(M_\lambda(D')=y)\right)^{\alpha-1}}$$

The first inequality is due to Lemma A.3, the second inequality is because $a_\lambda$ are non-negative, and the last inequality is because of (A.16) and the fact that both $\pi^+(\lambda)$ and $\pi(\lambda)$ are defined in $\mathbf{S}_{C,c}$, and thus their ratio is upper bounded by $\frac{C}{c}$ for any $\lambda$.

Now we only need to prove that for any fixed distribution $\pi$ that doesn't depend on value $y$, we have

$$\sum_{y \in \mathcal{Y}} \frac{\left(\sum_\lambda \pi(\lambda)\mathbb{P}(M_\lambda(D) = y)\right)^\alpha}{\left(\sum_\lambda \pi(\lambda)\mathbb{P}(M_\lambda(D') = y)\right)^{\alpha-1}} \le \sup_{\lambda \in \Lambda} e^{(\alpha-1)\mathrm{D}_\alpha\left(M_\lambda(D)\|M_\lambda(D')\right)}. \tag{A.17}$$

With this result, we immediately know the result holds for uniform distribution $\pi$ as a special case. To prove this result, we first observe that the function $f(u,v) = u^\alpha v^{1-\alpha}$ is a convex function. This is because the Hessian of $f$ is

$$\begin{pmatrix} \alpha(\alpha-1)u^{\alpha-2}v^{1-\alpha} & -\alpha(\alpha-1)u^{\alpha-1}v^{-\alpha} \\ -\alpha(\alpha-1)u^{\alpha-1}v^{-\alpha} & \alpha(\alpha-1)u^\alpha v^{-\alpha-1} \end{pmatrix},$$

which is easy to see to be positive semi-definite. And now, consider any distribution $\pi$, denote $u(\lambda) = \mathbb{P}(M_\lambda(D) = y)$ and $v(\lambda) = \mathbb{P}(M_\lambda(D') = y)$ by Jensen's inequality, we have

$$f\left(\sum_\lambda \pi(\lambda)u(\lambda), \sum_\lambda \pi(\lambda)v(\lambda)\right) \le \sum_\lambda \pi(\lambda)f(u(\lambda), v(\lambda)).$$

By adding the summation over $y$ on both side of the above inequality, we have

$$\sum_{y \in \mathcal{Y}} \frac{\left(\sum_\lambda \pi(\lambda)\mathbb{P}(M_\lambda(D) = y)\right)^\alpha}{\left(\sum_\lambda \pi(\lambda)\mathbb{P}(M_\lambda(D') = y)\right)^{\alpha-1}} \le \sum_{y \in \mathcal{Y}} \sum_\lambda \pi(\lambda)\frac{\mathbb{P}(M_\lambda(D) = y)^\alpha}{\mathbb{P}(M_\lambda(D') = y)^{\alpha-1}}$$

$$= \sum_\lambda \sum_{y \in \mathcal{Y}} \pi(\lambda)\frac{\mathbb{P}(M_\lambda(D) = y)^\alpha}{\mathbb{P}(M_\lambda(D') = y)^{\alpha-1}}$$

$$\le \sup_\lambda \sum_{y \in \mathcal{Y}} \frac{\mathbb{P}(M_\lambda(D) = y)^\alpha}{\mathbb{P}(M_\lambda(D') = y)^{\alpha-1}}.$$

The first equality is due to Fubini's theorem. And the second inequality is straight forward as one observe $\pi(\lambda)$ only depends on $\lambda$. This concludes the proof as we know that

$$e^{(\alpha-1)\mathrm{D}_\alpha(P^+\|P'^-)} \le \left(\frac{C}{c}\right)^\alpha \sup_\lambda \sum_{y \in \mathcal{Y}} \frac{\mathbb{P}(M_\lambda(D) = y)^\alpha}{\mathbb{P}(M_\lambda(D') = y)^{\alpha-1}}$$

$$= \left(\frac{C}{c}\right)^\alpha \sup_\lambda e^{(\alpha-1)\mathrm{D}_\alpha(M_\lambda(D)\|M_\lambda(D')}$$

or equivalently,

$$\mathrm{D}_\alpha(P^+\|P'^-) \le \frac{\alpha}{\alpha-1}\log\frac{C}{c} + \sup_{\lambda \in \Lambda}\mathrm{D}_\alpha\left(M_\lambda(D)\|M_\lambda(D')\right).$$

$\square$

We now present our crucial technical lemma for adaptive hyperparameter tuing with any distribution on the number of repetitions $T$. This is a generalization from [32].

**Lemma A.5.** *Fix $\alpha > 1$. Let $T$ be a random variable supported on $\mathbb{N}_{\ge 0}$. Let $f : [0,1] \to \mathbb{R}$ be the probability generating function of $K$, that is, $f(x) = \sum_{k=0}^\infty \mathbb{P}[T = k]x^k$.*

*Let $M_\lambda$ and $M'_\lambda$ be the base algorithm for $\lambda \in \Lambda$ on $\mathcal{Y}$ on $D$ and $D'$ respectively. Define $A_1 := \mathcal{A}(D, \pi^{(0)}, \mathcal{T}, C, c)$, and $A_2 := \mathcal{A}(D', \pi^{(0)}, \mathcal{T}, C, c)$. Then*

$$\mathrm{D}_\alpha(A_1\|A_2) \le \sup_\lambda \mathrm{D}_\alpha(M_\lambda\|M'_\lambda) + \frac{\alpha}{\alpha-1}\log\frac{C}{c} + \frac{1}{\alpha-1}\log\left(f'(q)^\alpha \cdot f'(q')^{1-\alpha}\right),$$

*where applying the same postprocessing to the bounding probabilities $P^+$ and $P'^-$ gives probabilities $q$ and $q'$ respectively. This means that, there exist a function set $g : \mathcal{Y} \to [0,1]$ such that $q = \mathbb{E}_{X \leftarrow P^+}[g(X)]$ and $q' = \mathbb{E}_{X' \leftarrow P'^-}[g(X')]$.*

*Proof of Lemma A.5.* We consider the event that $A_1$ outputs $y$. By definition, we have

$$A_1(y) = \sum_{k=1}^{\infty} \mathbb{P}(T=k)[\prod_{j=1}^{k} P_j(\leq y) - \prod_{i=1}^{k} P_j(< y)]$$

$$= \sum_{k=1}^{\infty} \mathbb{P}(T=k)[\sum_{i=1}^{k} P_i(y) \prod_{j=1}^{i-1} P_j(< y) \cdot \prod_{j=i+1}^{k} P_j(\leq y)]$$

$$\leq \sum_{k=1}^{\infty} \mathbb{P}(T=k)[\sum_{i=1}^{k} P^+(y) \prod_{j=1}^{i-1} P^+(< y) \cdot \prod_{j=i+1}^{k} P^+(\leq y)]$$

$$= \sum_{k=1}^{\infty} \mathbb{P}(T=k)[\sum_{i=1}^{k} P^+(y) \cdot P^+(< y)^{i-1} \cdot P^+(\leq y)^{k-i}]$$

$$= \sum_{k=1}^{\infty} \mathbb{P}(T=k)[P^+(\leq y)^k - P^+(< y)^k]$$

$$= f(P^+(\leq y)) - f(P^+(< y)) = P^+(y) \cdot \underset{X \leftarrow \text{Uniform}([P^+(<y),P^+(\leq y)])}{\mathbf{E}}[f'(X)].$$

The second equality is by partitioning on the events of the first time of getting $y$, we use $i$ to index such a time. The third inequality is using (A.7), (A.9), and (A.10). The third to last equality is by (A.12) and algebra. The second to last equality is by definition of the probability generating function $f$. The last equality follows from definition of integral.

Similarly, we have

$$A_2(y) \geq \sum_{k=1}^{\infty} \mathbb{P}(T=k)[P'^-(\leq y)^k - P'^-(< y)^k] = P'^-(y) \cdot \underset{X \leftarrow \text{Uniform}([P'^-(<y),P'^-(\leq y)])}{\mathbf{E}}[f'(X)].$$

The rest part of the proof is standard and follows similarly as in [32]. Specifically, we have

$$e^{(\alpha-1)\mathrm{D}_\alpha(A_1\|A_2)}$$

$$= \sum_{y \in \mathcal{Y}} A_1(y)^\alpha \cdot A_2(y)^{1-\alpha}$$

$$\leq \sum_{y \in \mathcal{Y}} P^+(y)^\alpha \cdot P'^-(y)^{1-\alpha} \cdot \underset{X \leftarrow [P^+(<y),P^+(\leq y)]}{\mathbf{E}}[f'(X)]^\alpha \cdot \underset{X' \leftarrow [P'^-(<y),P'^-(\leq y)]}{\mathbf{E}}[f'(X')]^{1-\alpha}$$

$$\leq \sum_{y \in \mathcal{Y}} P^+(y)^\alpha \cdot P'^-(y)^{1-\alpha} \cdot \underset{\substack{X \leftarrow [P^+(<y),P^+(\leq y)] \\ X' \leftarrow [P'^-(<y),P'^-(\leq y)]}}{\mathbf{E}}\left[f'(X)^\alpha \cdot f'(X')^{1-\alpha}\right]$$

$$\leq \left(\frac{C}{c}\right)^\alpha \sup_\lambda e^{(\alpha-1)\mathrm{D}_\alpha(M_\lambda(D)\|M_\lambda(D'))} \cdot \max_{y \in \mathcal{Y}} \underset{\substack{X \leftarrow [P^+(<y),P^+(\leq y)] \\ X' \leftarrow [P'^-(<y),P'^-(\leq y)]}}{\mathbf{E}}\left[f'(X)^\alpha \cdot f'(X')^{1-\alpha}\right].$$

The last inequality follows from Lemma A.4. The second inequality follows from the fact that, for any $\alpha \in \mathbb{R}$, the function $h : (0,\infty)^2 \to (0,\infty)$ given by $h(u,v) = u^\alpha \cdot v^{1-\alpha}$ is convex. Therefore, $\mathbb{E}[U]^\alpha \mathbb{E}[V]^{1-\alpha} = h(\mathbb{E}[(U,V)]) \leq \mathbb{E}[h(U,V)] = \mathbb{E}\left[U^\alpha \cdot V^{1-\alpha}\right]$ all positive random variables $(U,V)$. Note that $X$ and $X'$ are required to be uniform separately, but their joint distribution can be arbitrary. As in [32], we will couple them so that $\frac{X - P^+(<y)}{P^+(y)} = \frac{X' - P'^-(<y)}{P'^-(y)}$. In particular, this implies that, for each $y \in \mathcal{Y}$, there exists some $t \in [0,1]$ such that

$$\underset{\substack{X \leftarrow [P^+(<y),P^+(\leq y)] \\ X' \leftarrow [P'^-(<y),P'^-(\leq y)]}}{\mathbf{E}}\left[f'(X)^\alpha \cdot f'(X')^{1-\alpha}\right] \leq f'(P^+(< y)+t\cdot P^+(y))^\alpha \cdot f'\left(P'^-(< y) + t \cdot P'^-(y)\right)^{1-\alpha}$$

Therefore, we have

$$D_\alpha\left(A_1\|A_2\right) \leq \sup_\lambda D_\alpha\left(M_\lambda\|M_\lambda'\right) + \frac{\alpha}{\alpha-1}\log\frac{C}{c}$$

$$+ \frac{1}{\alpha-1}\log\left(\max_{\substack{y\in\mathcal{Y}\\t\in[0,1]}} f'(P^+(<y) + t\cdot P^+(y))^\alpha \cdot f'\left(P'^-(<y) + t\cdot P'^-(y)\right)^{1-\alpha}\right).$$

To prove the result, we simply fix $y_* \in \mathcal{Y}$ and $t_* \in [0,1]$ achieving the maximum above and define

$$g(y) := \left\{\begin{array}{ll} 1 & \text{if } y < y_* \\ t_* & \text{if } y = y_* \\ 0 & \text{if } y > y_* \end{array}\right.$$

The result directly follows by setting $q = \mathop{\mathbb{E}}_{X\leftarrow P^+}[g(X)]$ and $q' = \mathop{\mathbb{E}}_{X'\leftarrow P'^-}[g(X')]$. $\qquad\square$

Now we can prove Theorem 1, given the previous technical lemma. The proof share similarity to the proof of Theorem 2 in [32] with the key difference from the different form in Lemma A.5. We demonstrate this proof as follows for completeness.

*Proof of Theorem 1.* We first specify the probability generating function of the truncated negative binomial distribution

$$f(x) = \mathop{\mathbb{E}}_{T\sim\text{NegBin}(\theta,\gamma)}\left[x^T\right] = \left\{\begin{array}{ll} \frac{(1-(1-\gamma)x)^{-\theta}-1}{\gamma^{-\theta}-1} & \text{if } \theta \neq 0 \\ \frac{\log(1-(1-\gamma)x)}{\log(\gamma)} & \text{if } \theta = 0 \end{array}\right.$$

Therefore,

$$f'(x) = (1-(1-\gamma)x)^{-\theta-1} \cdot \left\{\begin{array}{ll} \frac{\theta\cdot(1-\gamma)}{\gamma^{-\theta}-1} & \text{if } \theta \neq 0 \\ \frac{1-\gamma}{\log(1/\gamma)} & \text{if } \theta = 0 \end{array}\right.$$

$$= (1-(1-\gamma)x)^{-\theta-1} \cdot \gamma^{\theta+1} \cdot \mathbb{E}[T].$$

By Lemma A.5, for appropriate values $q, q' \in [0,1]$ and for all $\alpha > 1$ and all $\hat\alpha > 1$, we have

$$D_\alpha\left(A_1 \| A_2\right)$$

$$\leq \sup_\lambda D_\alpha\left(M_\lambda \| M'_\lambda\right) + \frac{\alpha}{\alpha-1}\log\frac{C}{c} + \frac{1}{\alpha-1}\log\left(f'(q)^\alpha \cdot f'\left(q'\right)^{1-\alpha}\right)$$

$$\leq \varepsilon + \frac{\alpha}{\alpha-1}\log\frac{C}{c} + \frac{1}{\alpha-1}\log\left(\gamma^{\theta+1}\cdot\mathbb{E}[T]\cdot(1-(1-\gamma)q)^{-\alpha(\theta+1)}\cdot(1-(1-\gamma)q')^{-(1-\alpha)(\theta+1)}\right)$$

$$= \varepsilon + \frac{\alpha}{\alpha-1}\log\frac{C}{c}$$
$$\quad + \frac{1}{\alpha-1}\log\left(\gamma^{\theta+1}\cdot\mathbb{E}[T]\cdot\left((\gamma+(1-\gamma)(1-q))^{1-\hat\alpha}\cdot(\gamma+(1-\gamma)(1-q'))^{\hat\alpha}\right)^\nu\cdot(\gamma+(1-\gamma)(1-q))^u\right)$$

(Here, we let $\hat\alpha\nu = (\alpha-1)(1+\theta)$ and $(1-\hat\alpha)\nu + u = -\alpha(\theta+1)$)

$$\leq \varepsilon + \frac{\alpha}{\alpha-1}\log\frac{C}{c} + \frac{1}{\alpha-1}\log\left(\gamma^{\theta+1}\cdot\mathbb{E}[T]\cdot\left(\gamma+(1-\gamma)\cdot e^{(\hat\alpha-1)D_{\hat\alpha}\left(P^+\|P^-\right)}\right)^\nu\cdot(\gamma+(1-\gamma)(1-q))^u\right)$$

(Here, $1-q$ and $1-q'$ are postprocessings of some $P^+$ and $P'^-$ respectively and $e^{(\hat\alpha-1)D_{\hat\alpha}(\cdot\|\cdot)}$ is convex )

$$\leq \varepsilon + \frac{\alpha}{\alpha-1}\log\frac{C}{c} + \frac{1}{\alpha-1}\log\left(\gamma^{\theta+1}\cdot\mathbb{E}[T]\cdot\left(\gamma+(1-\gamma)\cdot e^{(\hat\alpha-1)\sup_\lambda D_{\hat\alpha}\left(M_\lambda\|M'_\lambda\right)+\hat\alpha\log\frac{C}{c}}\right)^\nu\cdot(\gamma+(1-\gamma)(1-q))^u\right)$$

(By Lemma A.4)

$$\leq \varepsilon + \frac{\alpha}{\alpha-1}\log\frac{C}{c} + \frac{1}{\alpha-1}\log\left(\gamma^{\theta+1}\cdot\mathbb{E}[T]\cdot\left(\gamma+(1-\gamma)\cdot e^{(\hat\alpha-1)\sup_\lambda D_{\hat\alpha}\left(M_\lambda\|M'_\lambda\right)+\hat\alpha\log\frac{C}{c}}\right)^\nu\cdot(\gamma+(1-\gamma)(1-q))^u\right)$$

$$\leq \varepsilon + \frac{\alpha}{\alpha-1}\log\frac{C}{c} + \frac{1}{\alpha-1}\log\left(\gamma^{\theta+1}\cdot\mathbb{E}[T]\cdot\left(\gamma+(1-\gamma)\cdot e^{(\hat\alpha-1)\sup_\lambda D_{\hat\alpha}\left(M_\lambda\|M'_\lambda\right)+\hat\alpha\log\frac{C}{c}}\right)^\nu\cdot\gamma^u\right)$$

(Here $\gamma \leq \gamma+(1-\gamma)(1-q)$ and $u \leq 0$)

$$= \varepsilon + \frac{\alpha}{\alpha-1}\log\frac{C}{c} + \frac{\nu}{\alpha-1}\log\left(\gamma+(1-\gamma)\cdot e^{(\hat\alpha-1)\sup_\lambda D_{\hat\alpha}\left(M_\lambda\|M'_\lambda\right)+\hat\alpha\log\frac{C}{c}}\right) + \frac{1}{\alpha-1}\log\left(\gamma^{\theta+1}\cdot\mathbb{E}[T]\cdot\gamma^u\right)$$

$$= \varepsilon + \frac{\alpha}{\alpha-1}\log\frac{C}{c} + \frac{\nu}{\alpha-1}\left((\hat\alpha-1)\sup_\lambda D_{\hat\alpha}\left(M_\lambda\|M'_\lambda\right)+\hat\alpha\log\frac{C}{c}\right.$$
$$\quad \left.+\log\left(1-\gamma\cdot\left(1-e^{-(\hat\alpha-1)\sup_\lambda D_{\hat\alpha}\left(M_\lambda\|M'_\lambda\right)+\hat\alpha\log\frac{C}{c}}\right)\right)\right) + \frac{1}{\alpha-1}\log\left(\gamma^{u+\theta+1}\cdot\mathbb{E}[T]\right)$$

$$= \varepsilon + \frac{\alpha}{\alpha-1}\log\frac{C}{c} + (1+\theta)\left(1-\frac{1}{\hat\alpha}\right)\sup_\lambda D_{\hat\alpha}\left(M_\lambda\|M'_\lambda\right) + (1+\theta)\log\frac{C}{c}$$
$$\quad + \frac{1+\theta}{\hat\alpha}\log\left(1-\gamma\cdot\left(1-e^{-(\hat\alpha-1)\sup_\lambda D_{\hat\alpha}\left(M_\lambda\|M'_\lambda\right)+\hat\alpha\log\frac{C}{c}}\right)\right) + \frac{\log(\mathbb{E}[T])}{\alpha-1} + \frac{1+\theta}{\hat\alpha}\log(1/\gamma)$$

(Here we have $\nu = \dfrac{(\alpha-1)(1+\theta)}{\hat\alpha}$ and $u = -(1+\theta)\left(\dfrac{\alpha-1}{\hat\alpha}+1\right)$)

$$= \varepsilon + \frac{\alpha}{\alpha-1}\log\frac{C}{c} + (1+\theta)\left(1-\frac{1}{\hat\alpha}\right)\sup_\lambda D_{\hat\alpha}\left(M_\lambda\|M'_\lambda\right) + (1+\theta)\log\frac{C}{c}$$
$$\quad + \frac{1+\theta}{\hat\alpha}\log\left(\frac{1}{\gamma}-1+e^{-(\hat\alpha-1)\sup_\lambda D_{\hat\alpha}\left(M_\lambda\|M'_\lambda\right)-\hat\alpha\log\frac{C}{c}}\right) + \frac{\log(\mathbb{E}[T])}{\alpha-1}$$

$$\leq \varepsilon + \frac{\alpha}{\alpha-1}\log\frac{C}{c} + (1+\theta)\left(1-\frac{1}{\hat\alpha}\right)\hat\varepsilon + (1+\theta)\log\frac{C}{c} + \frac{1+\theta}{\hat\alpha}\log\left(\frac{1}{\gamma}\right) + \frac{\log(\mathbb{E}[T])}{\alpha-1},$$

which completes the proof. $\qquad\square$

## B  Truncated Negative Binomial Distribution

We introduce the definition of truncated negative binomial distribution [32] in this section.

**Definition B.1.** (Truncated Negative Binomial Distribution [32]). Let $\gamma \in (0,1)$ and $\theta \in (-1,\infty)$. Define a distribution $\mathrm{NegBin}(\theta,\gamma)$ on $\mathbb{N}^+$ as follows:

- If $\theta \neq 0$ and $T$ is drawn from $\text{NegBin}(\theta, \gamma)$, then

$$\forall k \in \mathbb{N} \quad \mathbb{P}[T = k] = \frac{(1-\gamma)^k}{\gamma^{-\theta} - 1} \cdot \prod_{\ell=0}^{k-1} \left( \frac{\ell + \theta}{\ell + 1} \right)$$

  and $\mathbb{E}[T] = \frac{\theta \cdot (1-\gamma)}{\gamma \cdot (1-\gamma^\theta)}$. Note that when $\theta = 1$, it reduces to the geometric distribution with parameter $\gamma$.

- If $\theta = 0$ and $T$ is drawn from $\text{NegBin}(0, \gamma)$, then

$$\mathbb{P}[T = k] = \frac{(1-\gamma)^k}{k \cdot \log(1/\gamma)}$$

  and $\mathbb{E}[T] = \frac{1/\gamma - 1}{\log(1/\gamma)}$.

## C  Privatization of Sampling Distribution

### C.1  General Functional Projection Framework

In section 3.2, we define the projection onto a convex set $S_{C,c}$ as an optimization in terms of $\ell_2$ loss. More generally, we can perform the following general projection at the $j$-th iteration by considering an additional penalty term, with a constant $\nu$:

$$\min_f \; \|f - \pi^{(j)}\|_2 + \nu KL(\pi^{(j)}, f) \tag{C.1}$$

$$\text{s.t. } f \in S_{C,c}.$$

When $\nu = 0$, we recover the original $\ell_2$ projection. Moreover, it's worth noting that our formulation has implications for the information projection literature [9, 21]. Specifically, as the penalty term parameter $\nu$ approaches infinity, the optimization problem evolves into a minimization of KL divergence, recovering the objective function of information projection (in this instance, moment projection). However, the constraint sets in the literature of information projection are generally much simpler than our set $S_{C,c}$, making it infeasible to directly borrow methods from its field. To the best of our knowledge, our framework is the first to address this specific problem in functional projection and establish a connection to information projection in the DP community.

### C.2  Practical Implementation of Functional Projection

Optimization program (3.1) is essentially a functional programming since $f$ is a function on $\Lambda$. However, when $\Lambda$ represents a non-discrete parameter space, such functional minimization is typically difficult to solve analytically. Even within the literature of information projection, none of the methods considers our constraint set $S_{C,c}$, which can be viewed as the intersections of uncountable single-point constraints on $f$. To obtain a feasible solution to the optimization problem, we leverage the idea of discretization. Instead of viewing (3.1) as a functional projection problem, we manually discretize $\Lambda$ and solve (3.1) as a minimization problem over a discrete set. Note that such approximation is unavoidable in numerical computations since computers can only manage discrete functions, even when we solve the functional projection analytically. Moreover, we also have the freedom of choosing the discretization grid without incurring extra privacy loss since the privacy cost is independent of the size of parameter space. By converting $S_{C,c}$ into a set of finite constraints, we are able to solve the discrete optimization problem efficiently using CVXOPT [2].

## D  DP-HyPO with General Prior Distribution

In the main manuscript, we assume $\pi^{(0)}$ follows a uniform distribution over the parameter space $\Lambda$ for simplicity. In practice, informed priors can be used when we want to integrate knowledge about the parameter space into sampling distribution, which is common in the Bayesian optimization framework. We now present the general DP-HyPO framework under the informed prior distribution.

To begin with, we define the space of essentially bounded density functions with respect to $\pi^{(0)}$ as

$$S_{C,c}(\pi^{(0)}) = \{f \in \Lambda^{\mathbb{R}^+} : \text{ess sup } f/\pi^{(0)} \leq C, \text{ess inf } f/\pi^{(0)} \geq c, \int_{\alpha \in \Lambda} f(\alpha)\mathrm{d}\alpha = 1, f \ll \pi^{(0)}\}.$$

When $\pi^{(0)} = \frac{1}{\mu(\lambda)}$, we recover the original definition of $S_{C,c}$. Note that here $f \ll \pi^{(0)}$ means that $f$ is absolute continuous with respect to the prior distribution $\pi^{(0)}$ and this ensures that $S_{C,c}(\pi^{(0)})$ is non-empty. Note that such condition is automatically satisfied when $\pi^{(0)}$ is the uniform prior over the entire parameter space.

To define the projection of a density at the $j$-th iteration, $\pi^{(j)}$, into the space $S_{C,c}(\pi^{(0)})$, we consider the following functional programming problem:

$$\min_f \ \|f - \pi^{(j)}\|_2$$
$$\text{s.t.} \ \ f \in S_{C,c}(\pi^{(0)}),$$

which is a direct generalization of Equation (3.1). As before, $S_{C,c}(\pi^{(0)})$ is also convex and closed and the optimization program can be solved efficiently via discretization on $\Lambda$.

# E  Experiment Details

## E.1  MNIST Simulation

We now provide the detailed description of the experiment in Section 4.2.1. As specified therein, we consider two variable hyperparameters: the learning rate $\eta$ and clipping norm $R$, while keeping all the other hyperparameters fixed. We set the training batch size to be $256$, and the total number of epoch to be $10$. The value of $\sigma$ is determined based on the allocated $\varepsilon$ budget for each base algorithm. Specifically, $\sigma = 0.71$ for GP and $\sigma = 0.64$ for Uniform. For demonstration purposes, we set $C$ to 2 and $c$ to 0.75 in the GP method, so each base algorithm of Uniform has $\log C/c$ more privacy budget than base algorithms in GP method. In Algorithm 3, we set $\tau$ to 0.1 and $\beta$ to 1. To facilitate the implementation of both methods, we discretize the learning rates and clipping norms as specified in the following setting to allow simple implementation of sampling and projection for Uniform and GP methods.

**Setting E.1.** *we set a log-spaced grid discretization on $\eta$ in the range $[0.0001, 10]$ with a multiplicative factor of $\sqrt[3]{10}$, resulting in $16$ observations for $\eta$. We also set a linear-spaced grid discretization on $R$ in the range $[0.3, 6]$ with an increment of $0.3$, resulting in $20$ observations for $R$. This gives a total of $320$ hyperparameters over the search region.*

We specify the network structure we used in the simulation as below. It is the standard CNN in `Tensorflow Privacy` and `Opacus`.

```
class ConvNet(nn.Module):
    def __init__(self):
        super().__init__()
        self.conv1 = nn.Conv2d(1, 16, 8, 2, padding=3)
        self.conv2 = nn.Conv2d(16, 32, 4, 2)
        self.fc1 = nn.Linear(32 * 4 * 4, 32)
        self.fc2 = nn.Linear(32, 10)

    def forward(self, x):
        x = F.relu(self.conv1(x))
        x = F.max_pool2d(x, 2, 1)
        x = F.relu(self.conv2(x))
        x = F.max_pool2d(x, 2, 1)
        x = x.view(-1, 32 * 4 * 4)
        x = F.relu(self.fc1(x))
        x = self.fc2(x)
        return x
```

Despite the simple nature of MNIST, the simulation of training CNN with the two methods over each different fixed $T$ still take significant computation resources. Due to the constraints on computational resources, we conduct a semi-real simulation using the MNIST dataset. We cache the mean accuracy of $5$ independently trained models for each discretized hyperparameter and treat that as a proxy for the

"actual accuracy" of the hyperparameter. Each time we sample the accuracy of a hyperparameter, we add Gaussian noise with a standard deviation of $0.1$ to the cached mean. We evaluate the performance of the output model based on the "actual accuracy" corresponding to the selected hyperparameter.

## E.2  CIFAR-10 Simulation

We also provide a description of the experiment in Section 4.2.2. We set the training batch size to be $256$, and the total number of epoch to be $10$. The value of $\sigma$ is determined based on the allocated $\varepsilon$ budget for each base algorithm. Specifically, $\sigma = 0.65$ for GP and $\sigma = 0.6$ for Uniform. Regarding our GP method, we adopt the same set of hyperparameters as used in our MNIST experiments, which include $C = 2$, $c = 0.75$, $\tau = 0.1$, and $\beta = 1$. As usual, we discretize the learning rates and clipping norms as specified in the following Setting.

**Setting E.2.** *we set a log-spaced grid discretization on $\eta$ in the range $[0.0001, 1]$ with a multiplicative factor of $10^{0.1}$, resulting in $50$ observations for $\eta$. We also set a linear-spaced grid discretization on $R$ in the range $[0, 100]$ with an increment of $2$, resulting in $50$ observations for $R$. This gives a total of $2500$ hyperparameter combinations over the search region.*

We follow the same CNN model architecture with our MNIST experiments.

In Figure 2, we provide the hyperparameter landscape for $\sigma = 0.65$, as generated by BoTorch [3].

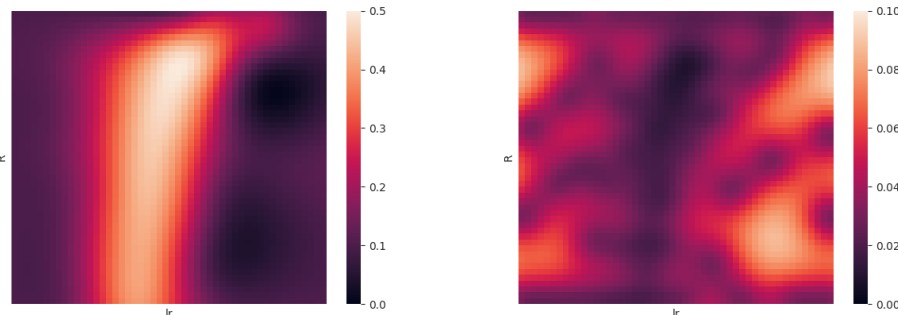

Figure 2: Mean and standard error of the accuracy of DP-SGD over the two hyperparameters for $\sigma = 0.65$. The learning rate (log-scale) ranges from $0.00001$ (left) to $1$ (right) while the clipping norm ranges from $0$ (top) to $100$ (bottom). The landscape for $\sigma = 0.6$ is similar, with a better accuracy.

## E.3  Federated Learning Simulation

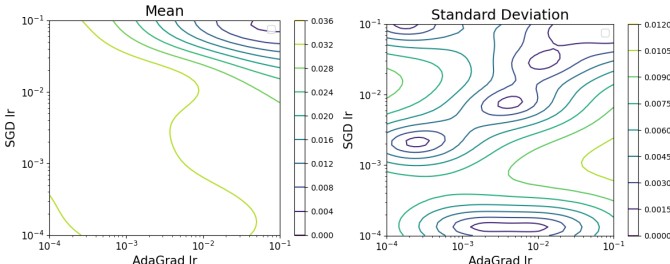

Figure 3: Mean and Standard Error of the loss of the FL over the two hyperparameters.

We now provide the detailed description of the experiment in Section 4.2.3. As specified therein, we considered a FL task on a proprietary dataset[5]. Our objective is to determine the optimal learning

---

[5]We are unable to report a lot of detail about the proprietary dataset due to confidentiality.

rates for the central server (using AdaGrad) and the individual users (using SGD). To simulate this scenario, we utilize the landscape generated by BoTorch [3], as illustrated in Figure 3, and consider it as our reference landscape for both mean and standard deviation of the loss for each hyperparameter. When the algorithm (GP or Uniform) visits a specific hyperparameter $\lambda$, our oracle returns a noisy score $q(\lambda)$ drawn from a normal distribution $N(\mu_\lambda, \sigma_\lambda)$. Figure 3 displays a heatmap that presents the mean ($\mu_\lambda$) and standard error ($\sigma_\lambda$) structure of the loss over these two hyperparameters, providing insights into the landscape's characteristics.

## F    Additional Related Work

In this section, we delve into a more detailed review of the pertinent literature.

We begin with non-private Hyperparameter Optimization, a critical topic in the realm of Automated Machine Learning (AutoML) [16]. The fundamental inquiry revolves around the generation of high-performing models within a specific search space. In historical context, two types of optimizations have proven significant in addressing this inquiry: architecture optimization and hyperparameter optimization. Architecture optimization pertains to model-specific parameters such as the number of neural network layers and their interconnectivity, while hyperparameter optimization concerns training-specific parameters, including the learning rate and minibatch size. In our paper, we incorporate both types of optimizations within our HPO framework. Practically speaking, $\Lambda$ can encompass various learning rates and network architectures for selection. For HPO, elementary methods include grid search and random search [22, 17, 15]. Progressing beyond non-adaptive random approaches, surrogate model-based optimization presents an adaptive method, leveraging information from preceding results to construct a surrogate model of the objective function [26, 41, 20, 30]. These methods predominantly employs Bayesian optimization techniques, including Gaussian process [33], Random Forest [18], and tree-structured Parzen estimator [5].

Another important topic in this paper is Differential Privacy (DP). DP offers a mathematically robust framework for measuring privacy leakage. A DP algorithm promises that an adversary with perfect information about the entire private dataset in use – except for a single individual – would find it hard to distinguish between its presence or absence based on the output of the algorithm [12].

Historically, DP machine learning research has overlooked the privacy cost associated with HPO [1, 39, 42]. The focus has only recently shifted to the "honest HPO" setting, where this cost is factored in [28]. Addressing this issue directly involves employing a composition-based analysis. If each training run of a hyperparameter upholds DP, then the overall HPO procedure adheres to DP through composition across all attempted hyperparameter values. A plethora of literature on the composition of DP mechanisms attempts to quantify a better DP guarantee of the composition. Vadhan et al. [36] demonstrated that though $(\varepsilon, \delta)$-DP possesses a simple mathematical form, deriving the precise privacy parameters of a composition is #-P hard. Despite this obstacle, numerous advanced techniques are available to calculate a reasonably accurate approximation of the privacy parameters, such as Moments Accountant [1], GDP Accountant [11], and Edgeworth Accountant [37]. The efficacy of these accountants is attributed to the fact that it is easier to reason about the privacy guarantees of compositions within the framework of Rényi differential privacy [27] or $f$-differential privacy [11]. These methods have found widespread application in DP machine learning. For instance, when training deep learning models, one of the most commonly adopted methods to ensure DP is via noisy stochastic gradient descent (noisy SGD) [4, 35], which uses Moments Accountant to better quantify the privacy guarantee.

Although using composition for HPO is a simple and straightforward approach, it carries with it a significant challenge. The privacy guarantee derived from composition accounting can be excessively loose, scaling polynomially with the number of runs. Chaudhuri et al. [7] were the first to enhance the DP bounds for HPO by introducing additional stability assumptions on the learning algorithms. [24] made significant progress in enhancing DP bounds for HPO without relying on any stability properties of the learning algorithms. They proposed a simple procedure where a hyperparameter was randomly selected from a uniform distribution for each training run. This selection process was repeated a random number of times according to a geometric distribution, and the best model obtained from these runs was outputted. They showed that this procedure satisfied $(3\varepsilon, 0)$-DP as long as each training run of a hyperparameter was $(\varepsilon, 0)$-DP. Building upon this, [32] extended the procedure to accommodate negative binomial or Poisson distributions for the repeated uniform selection. They also

offered more precise Rényi DP guarantees for this extended procedure. Furthermore, [8] explored a generalization of the procedure for top-$k$ selection, considering $(\varepsilon, \delta)$-DP guarantees.

In a related context, [28] explored a setting that appeared superficially similar to ours, as their title mentioned "adaptivity." However, their primary focus was on improving adaptive optimizers such as DP-Adam, which aimed to reduce the necessity of hyperparameter tuning, rather than the adaptive HPO discussed in this paper. Notably, in terms of privacy accounting, their approach only involved composing the privacy cost of each run without proposing any new method.

Another relevant area of research is DP selection, which encompasses well-known methods such as the exponential mechanism [25] and the sparse vector technique [13], along with subsequent studies. However, this line of research always assumes the existence of a low-sensitivity score function for each candidate, which is an unrealistic assumption for hyperparameter optimization.

