# OpenReview forum: "DP-HyPO: An Adaptive Private Framework for Hyperparameter Optimization"
_NeurIPS.cc/2023/Conference — NeurIPS 2023 poster_

### Official Review · Reviewer_o9LM · 2023-07-04

**Soundness:** 4 excellent
**Presentation:** 4 excellent
**Contribution:** 4 excellent
**Rating:** 7
**Confidence:** 3

**Summary:**

This paper presents DP-HyPO, a pioneering framework for adaptive private hyperparameter optimization. Privacy risks are often neglected in private ML model training. DP-HyPO employs a comprehensive differential privacy analysis to bridge the gap between private and non-private optimization methods (non-adaptive HPO method and adaptive HPO method). The framework's effectiveness is demonstrated through empirical evaluations on a diverse range of real-world and synthetic datasets. DP-HyPO offers a promising solution for enhancing model performance while preserving privacy.

**Strengths:**

1. The paper is written to a high standard and is easy to understand.
2. The availability of provided code facilitates reproducibility, making it a straightforward process.
3. Adaptive hyperparameter optimization algorithms have an important place in practice. DP-HyPO permits the flexible use of non-DP adaptive hyperparameter methods, it is a very meaningful research problem.



**Weaknesses:**

I only have one concern regarding this paper, which pertains to the utilization of MNIST datasets that are deemed to be overly simplistic and small in scale. The performance improvement of GP compared to the uniform sample is far less than 1 percent as shown in Figure 1.
As mentioned in the conclusion, I would also recommend incorporating a more challenging dataset or a search space with greater complexity in future iterations of your work.

**Questions:**

Why a more challenging experimental setup was not used at the very beginning of the experiment? I don't think it would be difficult to find a more appropriate scenario to perform the evaluations.

---

> ### Author Rebuttal · Authors · 2023-08-10
>
> We thank the reviewer for the constructive feedback!
>
> We totally agree that the current experiment settings like MNIST are indeed overly simplistic, and think that could be the underlying cause why our experiments have not shown a significant gain. To address this, we conducted new experiments on CIFAR10 (a much more challenging task), and revealed a **substantial disparity** between our proposed approach and the baseline. Please refer to our "general" rebuttal for more detail.

---

> > ### Comment · Reviewer_o9LM · 2023-08-16
> >
> > Thank you for the clarifying comments. It gives me further confidence to recommend that the paper is accepted.

---

> > > ### Author Response · Authors · 2023-08-16
> > >
> > > We appreciate the reviewer's carefully reading our rebuttal. Thanks again for the valuable feedback!

---

### Official Review · Reviewer_pfPa · 2023-07-05

**Soundness:** 2 fair
**Presentation:** 2 fair
**Contribution:** 2 fair
**Rating:** 4
**Confidence:** 3

**Summary:**

This paper introduces DP-HyPO, a framework for “adaptive” private hyperparameter optimization, aiming to bridge the gap between private and non-private hyperparameter optimization, that can allow practitioners to adapt to previous runs and focus on potentially superior hyperparameters. Besides, the arbitrary adaptive sampling distributions based on previous runs are allowed without any stability assumptions. The paper proposes experimental analysis to show the proposed methods' strengths.

**Strengths:**

1. The paper addresses the problem of hyperparam optimization, which is an important problem.
2. The paper proposes an interesting optimization framework, namely, DP-HyPO, which enables adaptive parameter selection under privacy constraints.
3. The paper provides sharp DP guarantees for adaptive private hyperparameter optimization.

**Weaknesses:**

1. The experiments are not abundant, further evaluation should be listed.  There are only two sections related to the experiments which I do not think can reveal the strengths of the proposed method. Additional in-deep analysis should be conducted.
2. In the experimental part, some related works are not compared. The author may want to add more related baselines listed in the Section of Related Works to show the efficiency.

**Questions:**

1.  What is the purpose of Figure 1?  I want to know the details of experimental settings.  Maybe I have lost some important information.
2.  What is the performance of other baselines? As the author states, some other baselines can be used for hyperparameter optimization.

**Limitations:**

The paper studies an important problem. However, the experimental section is unsatisfying and additional experimental analysis should be added. Besides, the author may want to clearly state the experimental settings.  Additionally, some non-adaptive methods should be compared such as Grid Search,  random search (RS), and Bayesian optimization (BO).

---

> ### Author Rebuttal · Authors · 2023-08-10
>
> We appreciate the reviewer for valuable feedback. While we respectfully acknowledge the input, we do have different viewpoints on certain aspects raised by the reviewer.
>
> **Empirical demonstration is important**: While our intention is to provide a rigorously theoretically-backed privacy accounting framework for HPO with DP consideration, we do believe a strong empirical demonstration is important. We acknowledge that the current experiment settings like MNIST are indeed overly simplistic, and think that could be the underlying cause why our experiments have not shown a significant gain. To address this, we conducted new experiments on CIFAR10 (a much more challenging task), and revealed a **substantial disparity** between our proposed approach and the baseline. Please refer to our "general" rebuttal for more detail.
>
> **Comparison to other related methods**: HPO with DP guarantee is an under-explored problem, and there are very limited related works.  The simple applications of methods like Grid Search and Random Search will result in the privacy cost growing linearly with the number of hyperparameter trials (proved in Liu & Talwar, 2019). Liu & Talwar (2019) and Papernot & Steinke (2021) proposed the uniform method (our baseline), and this is the only known algorithm whose DP guarantee is independent of the number of runs. Therefore, those methods (Grid search and Random search) are definitely undesirable when the hyperparameter space is non-trivial, which is the main focus of our paper.  This is the reason both us and the other two papers omit such comparison. We will make this more clear in the updated version.
>
> **The purpose of Figure 1**: Left panel of Figure 1 demonstrates the performance comparison of GP-instantiated DP-HyPO and the Uniform Method on MNIST; Right panel of Figure 1 demonstrates the same comparison on a real federated learning task with synthetically fitted loss landscape. We provided a detailed description in Appendix E of the original submission.
>
> Reference:
> [1] Nicolas Papernot and Thomas Steinke. Hyperparameter tuning with renyi differential privacy.
> In International Conference on Learning Representations, 2021.
> [2] Jingcheng Liu and Kunal Talwar. Private selection from private candidates. In Proceedings of the 51st Annual ACM SIGACT Symposium on Theory of Computing, pages 298–309, 2019.
> [3] Shubhankar Mohapatra, et al. "The role of adaptive optimizers for honest private hyperparameter selection." Proceedings of the aaai conference on artificial intelligence. Vol. 36. No. 7. 2022.

---

> ### Comment · Area_Chair_WNdY · 2023-08-19
> **please respond to authors' rebuttal**
>
> Dear reviewer,
>
>   The authors have responded to your review. Please read it and respond to them. Please note that other reviewers had different evaluations of this work. Have a look at their comments and see if it changes your opinion about this work.
>
> Thank you advance,
>
>   Your AC

---

### Official Review · Reviewer_sjks · 2023-07-05

**Soundness:** 2 fair
**Presentation:** 2 fair
**Contribution:** 3 good
**Rating:** 5
**Confidence:** 3

**Summary:**

Hyperparameter optimization (HPO) is an important step for enhancing the performance of private model training methods such as DP-SGD. Currently, most advanced HPO algorithms (e.g., Bayesian optimization) require to adaptively select the hyperparameters, but existing private HPO methods are non-adaptive. To fill in this gap, this paper proposed a differentially private adaptive HPO framework (DP-HyPO) which enables the conversion of any non-private HPO algorithm into a private one. The proposed DP-HyPO is shown to achieve a theoretical DP guarantee and outperform the non-adaptive DP-HPO baselines.


**Strengths:**

This work has tackled an important problem of private HPO. It contributes some new ideas for converting the non-DP adaptive HPO algorithm into a DP one without incurring substantial privacy cost. The proposed Framework 1 is simple but has good generalization ability.


**Weaknesses:**

1. The presentation of this paper needs to be improved. To be specific,

* The paper doesn't flow well. Some important information (e.g., definition of RDP) is referred to the appendix and other literature. I suggest to re-organize the paper, moving some less important information (e.g., Algorithm 2) into the appendix and including the key definition/concept in the main paper.

* Only showing the equations of the theoretical results is not enough, they need to be deeply analyzed for supporting the claims. For example, why is the base algorithm required to satisfy two RDP? Is this assumption realistic? Given the result of $\varepsilon'$, what's the insight for each term in it and why is it considered as a "sharp DP guarantee (Line 79)"?

2. This work has highlighted several times that handling infinite hyperparameter space is one of the key contributions. However, it is not clear how this is achieved. In Algorithm 3, the update rule of $\pi^{t+1}$ seems to be intractable for infinite $\Lambda$. In the experiments (Line 294), the hyperparameters are still discretized for the GP method.

3. Other minor issues:

* $\mathcal{T}$ and T are used inconsistently in the paper.

* The reference format needs to be carefully checked. The up-to-date paper should be cited instead of the arXiv version. For example, [20] has been published in ICLR. Some conference/paper titles should be capitalized (e.g., aaai -> AAAI, Dp-raft-> DP-RAFT, etc).

* The DP Gaussian process and DP Bayesian optimization works are related but not discussed.

**Questions:**

1. Line 203-204: It is claimed that "one can easily adapt the proof to other probability families?". This is a strong statement without any justification. Can you provide any example?

2. How is the constraint in Section 3.2 related to that in Framework 1? Is the method proposed in Section 3.2 only work when $\pi^{0}$ is uniform?

3. In Section 4.2.1, how is the budget $\varepsilon$ selected? How is the tradeoff between adaptivity and privacy (Lines 220-223) demonstrated in the empirical results?

**Limitations:**

The limitations of the simple empirical settings are discussed but left to future work.

---

> ### Author Rebuttal · Authors · 2023-08-10
>
> We appreciate the reviewer for valuable feedback.
>
> **Structure and flow**: We will restructure the paper to be of a better flow.
>
> **Advantage and practicality of Theorem 1**: The statement of Theorem 1 involves two RDP guarantees that **can be the same**, that is, we can have $\alpha = \hat{\alpha}$, and $\epsilon = \hat{\epsilon}$. Our results can improve the final privacy guarantee by optimizing the choice of $\alpha, \hat{\alpha}, \epsilon, \hat{\epsilon}$, but does not require the base algorithm to satisfy two different RDP guarantees. We believe this is realistic and provides more freedom for practitioners to obtain a better and sharp DP guarantee if the base algorithms happen to have more than one RDP guarantee. More importantly, most common algorithms in DP ML satisfy $(\alpha, \epsilon)$-RDP for any $\alpha>1$, including DP-SGD. This Theorem generalizes a similar theorem that is “sharp” for the Uniform method in Papernot and Steinke (2021) which relies on exactly the same assumptions. We will add more discussions in our next version.
>
> **Infinite hyperparameter space**: Theoretically, our framework allows the hyperparameter space $\Lambda$ to be of any topological structure, including being an infinite set with some continuous structure like a subset of $\mathbb{R}^p$. One benefit of this perspective is that it allows us to adaptively and iteratively choose proper discretization of the space as we need. Meanwhile, as pointed out by the reviewer, empirically we require discretization to make the convex optimization solvable. However, as compared to the previous work, we can tolerate a much finer level of discretization since the performance of the Uniform algorithm degrades a lot with the number of candidates increasing. We will make this point more clear in the updated version.
>
> **Adoption to other probability families**: The crucial observation here is that our results depend on the probability through its probability generating functions as in Lemma A.5 on line 513 in appendix. By using the corresponding pgf in the proof for the derivations in line 530, one can generalize our proof to other probability families. There are other probability families that are proved in Papernot & Steinke (2021), which we believe is easy to generalize as the pgfs are already computed and used in their proofs for uniform method. Although the idea is straightforward, it is loose to say this would be “easy” for an arbitrary family. We will revise this statement and make it strict in our next version.
>
> **General priors for DP-HyPO**: Our framework also works for the general priors other than uniform distribution. We spend two separate sections in Appendix C and D discussing this point. We limit the prior to be uniform distribution mostly due to its mathematical simplicity, but other general priors are also considered in our framework.
>
> **Selection of $\epsilon$**: In Section 4.2.1, the privacy budgets for both the GP-instantiated DP-HyPO and Uniform are selected to make a fair comparison between the two methods. In line 220-223, we discussed that a higher value of $\frac{C}{c}$ means more privacy budget spending on adaptivity. We demonstrate the effect of this trade-off mostly in Section 4.2.2, where we vary the value of $C$ (and set $c = \frac{1}{C}$). The detailed results can be found in Table 2. Our selection of $\epsilon$ for the base algorithms is very typical in the DP literature. For example, it’s similar to the selection in Abadi, et al. (2016).
>
>
> **Reference**:
>
> [1] Nicolas Papernot and Thomas Steinke. Hyperparameter tuning with renyi differential privacy.
> In International Conference on Learning Representations, 2021.
>
> [2] Jingcheng Liu and Kunal Talwar. Private selection from private candidates. In Proceedings of the 51st Annual ACM SIGACT Symposium on Theory of Computing, pages 298–309, 2019.
>
> [3] Martin Abadi, et al. "Deep learning with differential privacy." In Proceedings of the 2016 ACM SIGSAC conference on computer and communications security. 2016.

---

> > ### Comment · Reviewer_sjks · 2023-08-16
> >
> > I appreciate the authors for the detailed responses which have addressed most of my concerns. Therefore, I increase my score to 5 and hope the authors can revise the paper according to the responses, especially for the paper organization and clarifications of the theoretical results/claims.

---

> > > ### Author Response · Authors · 2023-08-16
> > > **Thanks!**
> > >
> > > We appreciate the reviewer's carefully reading our rebuttal. Will improve the paper according to your advice.

---

### Official Review · Reviewer_6uG2 · 2023-07-06

**Soundness:** 3 good
**Presentation:** 2 fair
**Contribution:** 3 good
**Rating:** 6
**Confidence:** 2

**Summary:**

In the paper "DP-HyPO: An Adaptive Private Framework for Hyperparameter Optimization" the authors propose a framework for differential privacy HPO turning adaptive aka. model-based HPO methods into privacy preserving HPO methods. In their empirical study, the authors demonstrate that despite restrictions and cuts for achieving DP, the adaptive HPO method involving a GP is still performing consistently better compared to uniformly sampling.

**Strengths:**

+ Flexible framework that allows to turn commonly used model-based HPO methods into DP-HPO methods
+ Improvement over uniform baseline
+ Theoretically guarantees for maintaining DP

**Weaknesses:**

- I am not an expert in DP but I think that privacy concerns should definitely be considered also in HPO. However, it seems like the framework is mostly centered around deep learning methods. At least there are certain assumptions for the guarantees to hold, e.g., that an ML algorithm run can be repeated to achieve a certain degree of privacy and also the experiments are solely limited to deep learning scenarios. As HPO methods are typically used in a much broader sense, it is therefore questionable to what extent the proposed framework also generalizes to other learning algorithms/models.

- The common HPO community might not be familiar with the term "privacy cost" and it could make sense to at least briefly explain what the auhtors mean by privacy cost.

# Minor
l. 82: "gurantees"
l. 168: "bridges"
l. 198 "proportion"
l. 227 "update"
l. 228f the sentence is broken, maybe a word missing?

**Questions:**

- What about deterministic learning algorithms? Is DP also preserved via the framework?
- Could not model-free HPO methods like Hyperband be used to maintain differential privacy? Since Hyperband is a relatively strong HPO method, this could potentially be another argument to stick to such approaches without using a model.

**Limitations:**

A discussion of limitations is missing in the paper.

---

> ### Author Rebuttal · Authors · 2023-08-10
>
> We thank the reviewer for the constructive feedback!
>
> **For general ML problems**: We clarify that our results hold for much broader HPO problems beyond deep learning. The requirement of “an ML algorithm run can be repeated to achieve a certain degree of privacy” is not an assumption of our framework. Although our framework runs an iterative meta-algorithm that has a similar structure like deep learning, we do not limit the base algorithms to be deep learning. Actually, our framework is beneficial to any DP algorithm which has hyperparameters. For example, the sparse vector technique and propose-test-release (Cynthia & Roth, 2014) are two fundamental DP algorithms whose performance largely depends on the choice of hyperparameters.
>
> **Minors**:
> We appreciate the reviewer's careful reading of our paper. We will correct all those good catches in the updated version. We will also make the meaning of private cost more clear.
>
> **DP and deterministic algorithms**: The notion of differential privacy (Dwork et al., 2006) is a probabilistic guarantee, which inherently requires the algorithm to be randomized. That being said, any nontrivial deterministic algorithm does not satisfy DP.
>
> **DP-HyPO with Hyperband**: Thanks for the advice. As we discussed in the paper, Gaussian Process is only one instantiation of the framework. Hyperband, as a bandit-based approach to HPO, should be suited with our DP-HyPO framework. We will list exploring this method as a future direction.
>
> [1] Dwork, Cynthia, and Aaron Roth. "The algorithmic foundations of differential privacy." Foundations and Trends® in Theoretical Computer Science 9.3–4 (2014): 211-407
>
> [2] Cynthia Dwork, Frank McSherry, Kobbi Nissim, and Adam Smith. Calibrating noise to 375 sensitivity in private data analysis. In Theory of Cryptography: Third Theory of Cryptography 376 Conference, TCC 2006, New York, NY, USA, March 4-7, 2006. Proceedings 3, pages 265–284. 377 Springer, 2006.

---

> ### Comment · Area_Chair_WNdY · 2023-08-19
> **Respond to authors**
>
> Dear reviewer,
>
>   The authors have responded to your comments. Please read their rebuttal and respond to them with whether their comments addressed your concerns.
>
> Thank you in advance,
>
>   Your AC

---

### Official Review · Reviewer_11th · 2023-07-13

**Soundness:** 3 good
**Presentation:** 4 excellent
**Contribution:** 3 good
**Rating:** 6
**Confidence:** 4

**Summary:**

This work proposes a differentially private (DP) adaptive hyperparameter optimization algorithm called DP-HyPO, which encompasses several existing DP non-adaptive hyperparameter optimization algorithms. DP-HyPO is able to deal with hyperparameters that come with infinitely many values and by leveraging the previous output from the base algorithm, DP-HyPO outperforms the non-adaptive algorithms. The paper gives a privacy analysis of the proposed DP-HyPO and showcases two practical applications of private hyperparameter tuning by instantiating the DP-HyPO framework with Gaussian Process (GP).

**Strengths:**

Clear and intriguing presentation. The problem and several design choices are well motivated.

New theoretical result analyzing the privacy loss of an adaptive private hyperparameter optimization algorithm.


**Weaknesses:**

Experiment results are not appealing enough. They do not suggest a significant advantage of DP-HyPO compared to the existing non-adaptive algorithms.

**Questions:**

How can one interpret $\hat{\pi}^{(0)} = \pi^{(0)} \cdot \mu(\Lambda)$ in Framework 1, which is a product of two distributions?

What is the run time of performing the projection in Eq.(3.1)? Since one needs to find the optimal function in $S_{C, c}$ at every iteration of DP-HyPO, a long computation time might limit the practicality of DP-HyPO.

In Section 4, is it possible to derive convergence guarantees (i.e., the utility of the private algorithm) for DP-HyPO with Gaussian Process?

In Section 4.2.1 MINST simulation, how does DP-HyPO perform in the low privacy regime, say, each base algorithm has a privacy loss $\epsilon = 0.1$ (which is also practical), and the total privacy loss of DP-HyPO needs to be $\epsilon = 1$?

 In Section 4.2.2 applying DP-HyPO to federated learning, what is the privacy loss $\epsilon$ here?


**Limitations:**

Some limitations of this work were discussed in the conclusion.
Broader impact was not discussed.

---

> ### Author Rebuttal · Authors · 2023-08-10
>
> We thank the reviewer for the constructive feedback!
>
> **MNIST is simplistic**: We acknowledge that the current experiment settings like MNIST are indeed overly simplistic, and hypothesize that could be the underlying cause why our experiments have not shown a significant gain. To address this, we conducted new experiments on CIFAR10, and revealed a **substantial disparity** between our proposed approach and the baseline. Please refer to our "general" rebuttal for more detail.
>
> **Definition of $\mu(\Lambda)$**: Here, $\mu(\Lambda)$ is the total measure of the hyperparameter space, which is a real number, and it is a scalar multiplication. We will make this clearer in the writing.
>
> **Run time of projection in Eq (3.1)**: The projection is a convex optimization problem over the space of $\Lambda$. We treat this as an abstract and separate subproblem in the theoretical part of the paper, as it requires the specification and the discretization of $\Lambda$. In our experiment part, we use CVX to solve this convex problem and the runtime of any discretization is negligible compared to the base algorithm.
>
> **The utility guarantee of DP-HyPO with Gaussian Process**: We agree this is a very important problem. However, it is also hard to obtain meaningful utility guarantees. In the previous paper by Liu & Talwar (2019), they provided several utility guarantees for different instantiations of the uniform method. The only relevant result to our algorithm is Theorem 3.3 therein, which proves the the (iterated) uniform method can achieve at least as good as one-iteration of uniform selection with high probability. In the other paper by Papernot & Steinke (2021), no utility result is presented. We believe useful utility results of DP-HyPO are even more challenging than results of uniform methods, and the specification to Gaussian Process requires in-depth analysis of the selection as well as the GP itself. We will list it as an important future direction.
>
> **The performance for MNIST when $\epsilon = 0.1$**: This is actually the high privacy regime. Training deep learning model from scratch with this small privacy budget is super challenging. Empirically, $\epsilon$ should be at least 2 to have meaningfully performance for deep learning tasks (for example, De et al. (2023) shows a state-of-art accuracy of 60% for CIFAR 10 when $\epsilon = 1$, as compared to >95% in the non-private case).
>
> **Section 4.2.2**: In this section, we present a synthetic experiment where we use the **same** loss landscape from a practical federated learning task (Figure 2) for both DP-HyPO and uniform method. This reflects the scenario where DP-Hypo has a larger privacy budget than the uniform algorithm. In this case, we experiment on different choices of $C$ to see the benefit of using the extra privacy budget for better adaptivity. The results in Table 2 demonstrate a larger $C$ means a lower loss.
>
> **Reference**:
>
> [1] Jingcheng Liu and Kunal Talwar. Private selection from private candidates. In Proceedings of the 51st Annual ACM SIGACT Symposium on Theory of Computing, pages 298–309, 2019.
>
> [2] Nicolas Papernot and Thomas Steinke. Hyperparameter tuning with renyi differential privacy.
> In International Conference on Learning Representations, 2021.
>
> [3] De, Soham, et al. "Unlocking high-accuracy differentially private image classification through scale." arXiv preprint arXiv:2204.13650 (2022).

---

> > ### Comment · Reviewer_11th · 2023-08-16
> >
> > Thank you very much for the detailed response!

---

### Author Rebuttal · Authors · 2023-08-10

We acknowledge that the current experiment settings like MNIST are indeed overly simplistic, and think that could be the underlying cause why our experiments have not shown a significant gain. To address this, we conducted new experiments on CIFAR10 (a much more challenging task), and revealed a substantial disparity (>2% performance improvement) between our proposed approach and the baseline. Please refer to our PDF for new experiment results.

---

### Decision · Program_Chairs · 2023-09-21

**Decision:**

Accept (poster)

**Comment:**

Most studies so far about private-ML focused either on the training phase or the inference phase. This paper looks at hyper-parameters tuning and develops a differential private way to complete this task. By careful analysis, the authors bring us closer to being able to run ML pipelines while preserving privacy.